# MicroRNAs in tear fluids predict underlying molecular changes associated with Alzheimer's disease

Printha Wijesinghe[1], Jeanne Xi[1], Jing Cui[1], Matthew Campbell[1], Wellington Pham[2,3], Joanne A Matsubara[1,4]

Extracellular circulating microRNAs (miRNAs) have been discussed as potential biomarkers for Alzheimer's disease (AD) diagnosis. As the retina is a part of the CNS, we hypothesize that miRNAs expression levels in the brain, particularly neocortex–hippocampus, eye tissues, and tear fluids are similar at different stages of AD progression. Ten miRNA candidates were systematically investigated in transgenic APP-PS1 mice, noncarrier siblings, and C57BL/6J wild-type controls at young and old ages. Relative expression levels of tested miRNAs revealed a similar pattern in both APP-PS1 mice and noncarrier siblings when compared with age- and sex-matched wild-type controls. However, the differences seen in expression levels between APP-PS1 mice and noncarrier siblings could possibly have resulted from underlying molecular etiology of AD. Importantly, miRNAs associated with amyloid beta (Aβ) production (-101a, -15a, and -342) and proinflammation (-125b, -146a, and -34a) showed significant up-regulations in the tear fluids with disease progression, as tracked by cortical Aβ load and reactive astrogliosis. Overall, for the first time, the translational potential of up-regulated tear fluid miRNAs associated with AD pathogenesis was comprehensively demonstrated.

## Introduction

Sporadic Alzheimer's disease (AD) contributes 60–70% of all dementias (World Health Organization, 2012). AD is defined by two pathological hallmarks: amyloid beta (Aβ) plaques and τ neurofibrillary tangles (Montine et al, 2012). Autopsy-based neuropathological diagnosis is the gold standard in AD research. There is often a discrepancy between clinical and pathology-based AD diagnoses, which is related to the prevalence of non-AD dementias, including alpha (α)- synucleinopathies, non-AD tauopathies, hippocampal sclerosis, and many types of cerebrovascular disease, all of which can mimic AD clinically (Dickson, 2009; Nelson et al, 2012; Wijesinghe et al, 2016a, 2016b). Genome-wide association studies, epigenetic

evaluations, such as miRNA–RNA linkage or association mapping for AD-relevant neurological pathways, appear as useful diagnostic approaches because it has recently become apparent that miRNA-mediated mRNA-targeted regulatory mechanisms involve a large number of pathogenic and highly integrated gene expression pathways in the central nervous system (CNS) (Zhao et al, 2015a, 2015b; Liu et al, 2018).

miRNAs are endogenous, noncoding single stranded, ~21–23 nucleotides length, small RNA molecules, and comprise a large family of post-transcriptional regulators of gene expression. About half of the all currently identified miRNAs are intragenic and processed mostly from introns and relatively few exons of protein-coding genes, whereas the remaining are intergenic, transcribed independently of a host gene, and regulated by their own promoters (O'Brien et al, 2018). In most cases, miRNAs interact with the 3′ UTR of target mRNAs to induce mRNA degradation and translational repression. miRNAs can be secreted extracellularly and served as biomarkers for diseases, and in addition may play important roles in intercellular communications (O'Brien et al, 2018). Because of their association with cell-derived nanovesicles (e.g., exosomes), RNA-binding proteins (e.g., Argonaute 2), or high-density lipoproteins, extracellular miRNAs are found to be stable (Creemers et al, 2012). Deregulated miRNAs have been identified under several human pathological conditions such as cancers, cardiovascular diseases, neurological disorders, autoimmune diseases, or in inflammatory states (Dolati et al, 2018; Nunez et al, 2020; Wei et al, 2020).

In the CNS, miRNAs act as key regulators of functions such as neurite outgrowth, dendritic spine morphology, neuronal differentiation, and synaptic plasticity (Cao et al, 2016; Hu & Li, 2017; Wijesinghe et al, 2022). About 2,650 different miRNA expressions have been reported in human tissues and only about 45 of these are abundant in the brain and retina (Pogue et al, 2011; Zhao et al, 2015a, 2015b; Hill & Lukiw, 2016). Pogue and Lukiw study (2018) reported seven miRNAs (-7, -9-1, -23a/-27a, -34a, -125b-1, -146a, and -155) that were significantly increased in abundance in AD-affected superior temporal lobe neocortex (Brodmann A22) and in the degenerated retina. Pogue and Lukiw suggested that these miRNAs are involved in the coordination of Aβ production and clearance,

[1]Department of Ophthalmology & Visual Sciences, Faculty of Medicine, The University of British Columbia, Eye Care Centre, Vancouver, Canada   [2]Department of Radiology and Radiological Sciences, Vanderbilt University Medical Centre, Nashville, TN, USA   [3]Vanderbilt University Institute of Imaging Science, Vanderbilt University Medical Center, Nashville, TN, USA   [4]Djavad Mowafaghian Centre for Brain Health, The University of British Columbia, Vancouver, Canada

Correspondence: jms@mail.ubc.ca

phagocytosis, innate-immune, pro-inflammatory, and neurotrophic signaling and/or synaptogenesis in diseased tissues. In the past decade, there is an enormous number of studies that have reported deregulated single miRNA or panel of miRNAs in early or severe AD-affected human postmortem brain specimens; AD patients' cerebrospinal fluids (CSFs), serum, and plasma samples; and samples obtained from different transgenic AD animal models or cell lines. However, no consensus has been achieved across the studies.

Despite accumulated evidence on miRNA deregulation in AD, lacking standardization in samples, sampling time points and quantification methods will impede the chance to characterize miRNAs as potential biomarkers for AD. Recently, Kenny et al (2019) demonstrated differentially expressed protein and miRNA profiles in the tear fluids of AD patients. Tear fluids are potential eye samples that can be obtained noninvasively and is a logical choice as they are relatively easy to sample and can be collected longitudinally at different disease stages and frozen for comparative studies. Moreover, tear fluids appear as a best choice for biomarker discovery over blood samples as the haemolysis during sample preparation alters miRNA content in blood samples (Kirschner et al, 2011). As part of the CNS, cellular processes that occur in the brain also occur in the retina. Several proposed mechanisms have been suggested as retinal changes because of AD, such as neurodegeneration, inflammation, amyloid misfolding, and amyloid angiopathy (Ning et al, 2008; Koronyo et al, 2017; Lee et al, 2020). Therefore, in this study, we hypothesize that the brain (neocortex–hippocampus), eye (retina), and tear fluids will share similarities in their miRNAs expression levels at different stages of AD progression. If proven correct, our results would support the use of tear samples to assess molecular changes in the CNS of individuals with AD. Here, we explore the translational potential of deregulated tear fluid miRNAs in the pathogenesis of AD to facilitate future tear fluid miRNA-based assays for diagnostic and prognostic purposes.

# Results

Of the 10 tested miRNAs, novel miRNA candidates -302c and -653 did not express at Ct <40 in the tested samples of all three mice groups. Remaining eight miRNAs' expression levels were determined across the samples at young and old ages, with ageing, and/or disease progression. Disease progression was confirmed in the neocortex–hippocampus of old Tg mice based on extracellular Aβ deposits or plaques surrounded by reactive astrocytes (Fig 1B).

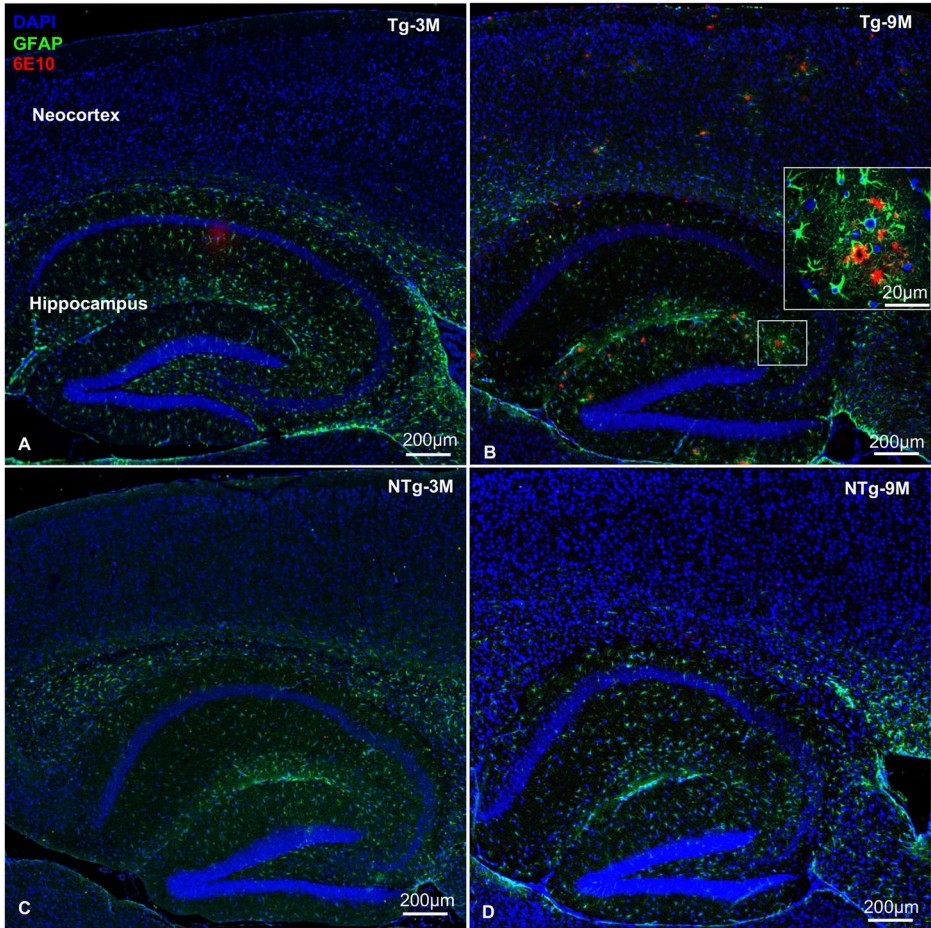

**Figure 1. Colocalization of Aβ plaques surrounded by reactive astrocytes in the neocortex–hippocampus.**
**(A, B, C, D)** Representative images obtained for (A) young (n = 4 per group, 12–16 wk, females) and (B) old (n = 4 per group, 36–40 wk, females) Tg mice are compared with matched (C) young and (D) old NTg siblings. Antibodies 6E10 and GFAP were used to localize the Aβ plaques and astrocytes, respectively, using double immunofluorescence staining. DAPI was used for nuclei staining. **(B)** Higher magnification of Aβ plaques surrounded by reactive astrocytes is shown in panel (B). Scale bars: 200 μm and 20 μm. (Aβ, amyloid beta; Tg, transgenic APP-PS1 mice; NTg, noncarrier siblings; GFAP, glial fibrillary acidic protein)

These typical AD-related neuropathological changes were obviously minimal or absent in young Tg mice and young and old NTg siblings (Fig 1A, C, and D). In addition, we also determined the relative mRNA expression levels for two human transgenes *APP* and *PSEN1* and endogenous mouse genes *App* and *Psen1* at young and old ages. All three mice groups were investigated to see their expression pattern between neocortex–hippocampus and eye tissues (Table 1 and Fig 2). Our gene expression data in eye tissues (Fig 2A and E) and neocortex–hippocampus (Fig 2C and G) demonstrated a very strong level of expression of human *PSEN1* mRNA in Tg mice ($P < 0.01$, log$_2$[Fold Change or FC] range: 10.9–20.0) and almost same level of expression of human *APP* mRNA ($P > 0.05$) in Tg mice, NTg siblings, and WT controls. Similarly, our gene expression data in eye tissues (Fig 2B and F) and neocortex–hippocampus (Fig 2D and H) demonstrated a significant increase in mouse *App* mRNA level in Tg mice ($P < 0.05$, log$_2$[FC] range: 1.1–1.9) and almost same level of expression of mouse *Psen1* mRNA ($P > 0.05$) in Tg mice, NTg siblings, and WT controls.

## miRNAs relative expression levels in neocortex–hippocampus, eye tissues, and tear fluids

### Tg mice versus NTg siblings

The expression levels of eight miRNAs in neocortex–hippocampus (Fig 3A), eye tissues (Fig 3B), and tear fluids (Fig 3C) were compared between Tg mice and NTg siblings (Table 2 and Fig 3). In young Tg mice, miRNAs -125b, -140, -146a, -15a, -342, -34a, and -374c in neocortex–hippocampus, -140 in eye tissues, and -34a in tear fluids showed significant up-regulations; and miRNAs -125b, -146a, and -374c in eye tissues showed significant down-regulations. In old Tg, miRNAs -140, -15a, and -374c in neocortex–hippocampus, -146a, -342, and -374c in eye tissues and -101a, -125b, -146a, -34a, and -374c in tear fluids showed significant up-regultions; and miRNAs -146a and -342 in neocortex–hippocampus showed significant down-regulations. Overall, Tg versus NTg comparision revealed a similar trend in miRNAs expression levels between neocortex–hippocampus and tear fluids in Tg mice at young age.

### Tg mice versus WT controls

The expression levels of eight miRNAs in neocortex–hippocampus (Fig 4A), eye tissues (Fig 4B), and tear fluids (Fig 4C) were compared between Tg mice and WT controls (Table 3 and Fig 4). In young Tg mice, most of the tested miRNAs showed significant or a trend toward down-regulation in all three samples. Particularly, miRNAs -101a and -15a in neocortex–hippocampus, -125b, -342, and -34a in eye tissues, and -101a, -15a, -342, -34a, and -374c in tear fluids were significantly down-regulated. In the old Tg, most of them showed significant or a trend toward up-regulation in all three samples. Paticularly, all tested miRNAs showed significant up-regulation in tear fluids. Except -342, remaining all miRNAs showed significant up-regulation in eye tisses. Whereas, miRNAs -125b, -140, -146a, -15a, and -374c showed significant up-regulations and -101a and -342 showed significant down-regulations in neocortex–hippocampus. Overall, Tg versus WT comparison showed a pattern where miRNAs expression levels were generally similar across neocortex–hippocampus, eye tissues, and tear fluids at young and old ages.

### NTg siblings versus WT controls

As per manufacturer, NTg siblings are typically made from same genetic background except two inserted transgenes which express a chimeric mouse/human amyloid precursor protein (Mo/Hu*APP*695swe) and a mutant human presenilin 1 (*PSEN1*-dE9) in Tg mice. Thus, we compared NTg sibling versus C57BL/6J WT control, which is the most widely used background strain in transgenic mouse models to see the difference between these two controls. Tested miRNAs' relative expression levels were illustrated for neocortex–hippocampus (Fig 5A), eye tissues (Fig 5B), and tear fluids (Fig 5C) (Table 4). In young NTg, most of the tested miRNAs showed significant or a trend toward down-regulation in all three samples. Particularly, miRNAs -101a, -146a, -15a, -342, -34a, and -374c in neocortex–hippocampus, -101a, -140, -342, and -34a in eye tissues, and excluding -146a remaining seven miRNAs in tear fluids showed significant down-regulations. In old NTg, most of the tested miRNAs showed significant or a trend toward up-regulation in all

**Table 1.  Relative mRNA expression level in neocortex–hippocampus and eye tissues of young (12–16 wk, n = 4 per group, females) and old (36–40 wk, n = 4 per group, females) Tg mice were compared with respective matched NTg siblings and WT controls.**

| Genes | Relative mRNA levels (Log$_2$[FC] >1.0 or < −1.0 and $P < 0.05$) | | | | | | | |
|---|---|---|---|---|---|---|---|---|
| | Tg versus NTg (young) | | Tg versus NTg (old) | | Tg versus WT (young) | | Tg versus WT (old) | |
| | N&H | Eye | N&H | Eye | N&H | Eye | N&H | Eye |
| Hu*APP* | −0.65 | −0.09 | **−1.13** | 0.04 | −0.95 | 0.91 | 0.93 | 0.81 |
| | (0.163) | (0.588) | **(0.001)** | (0.575) | (0.066) | (0.018) | (0.019) | (0.005) |
| Hu*PSEN1* | **10.95** | **19.97** | **15.81** | **19.87** | **18.26** | **16.06** | **20.03** | **16.45** |
| | **(0.002)** | **(0.002)** | **(0.003)** | **(<0.001)** | **(0.002)** | **(0.002)** | **(0.003)** | **(<0.001)** |
| Mo*App* | **1.1** | **1.29** | **1.39** | **1.67** | **1.91** | **1.56** | **1.38** | **1.54** |
| | **(0.023)** | **(0.001)** | **(0.007)** | **(0.009)** | **(0.014)** | **(0.003)** | **(0.006)** | **(0.01)** |
| Mo*Psen1* | 0.58 | 0.14 | 0.58 | 0.41 | 0.78 | 0.32 | 0.48 | 0.11 |
| | (0.007) | (0.118) | (0.008) | (0.021) | (0.013) | (0.012) | (0.015) | (0.21) |

Differentially expressed mRNAs are indicated (bold) based on $P < 0.05$ (Welch's $t$ test, two-tailed) and log$_2$[FC] >1.0 considered up-regulated (black color) and log$_2$[FC] <−1.0 considered down-regulated (blue color). (N&H, neocortex–hippocampus; FC, fold change)

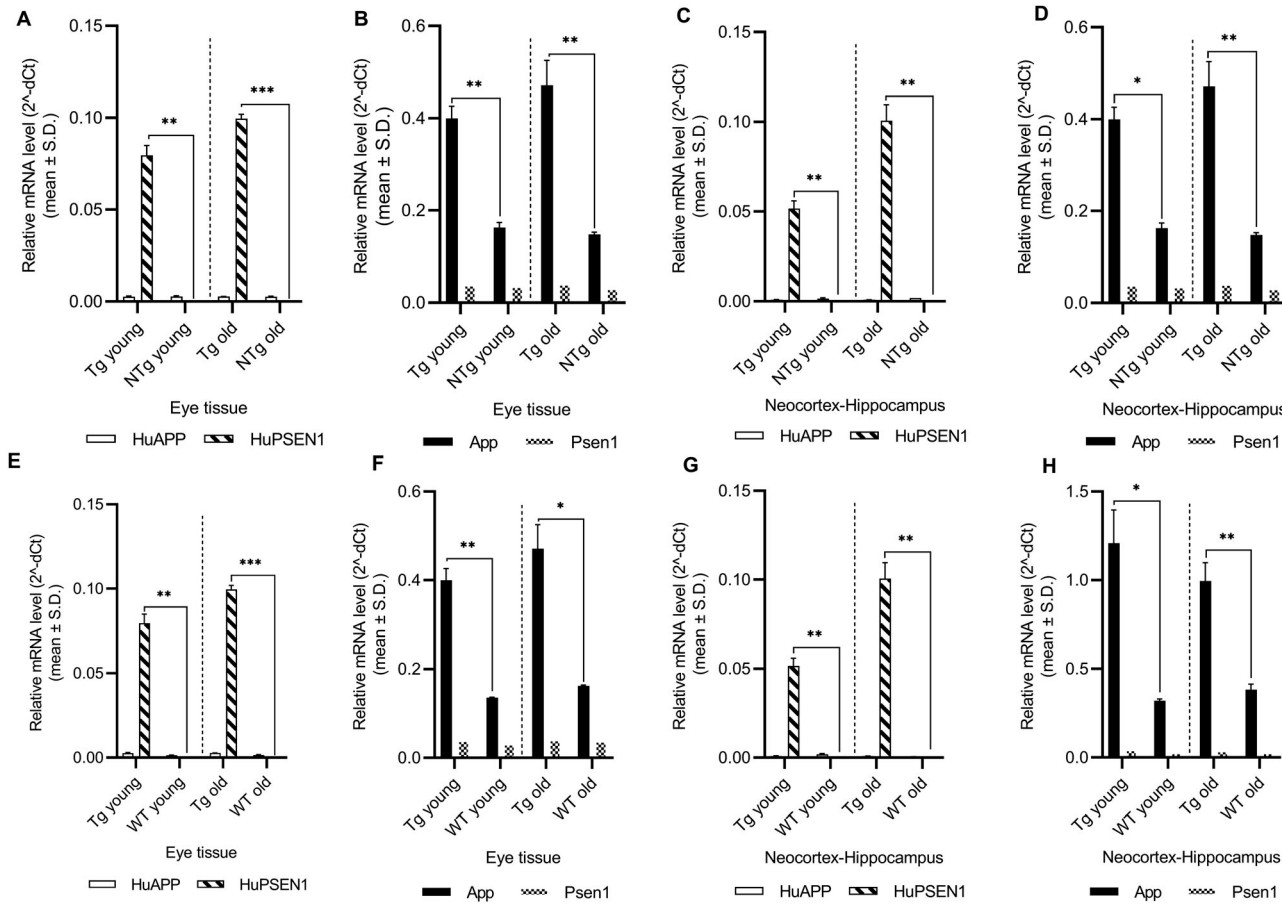

**Figure 2. Relative expression levels of two human transgenes *APP* and *PSEN1* and endogenous mouse genes *App* and *Psen1*.**
**(A, B, C, D, E, F, G, H)** mRNAs relative expression levels (mean ± Std. $2^{-\Delta Ct}$) in (A, B, E, F) eye tissues and (C, D, G, H) neocortex–hippocampus of young (n = 4 per group, 12–16 wk, females) and old (n = 4 per group, 36–40 wk, females) Tg mice, NTg siblings, and C57BL/6J WT controls are illustrated. Differentially expressed mRNAs are indicated for each comparison based on two-fold intergroup difference (>2.0 or <0.5) and *P*-value as *$P < 0.05$, **$P < 0.01$, ***$P < 0.001$ (Welch's *t* test). (Tg, transgenic APP-PS1 mice; NTg, noncarrier siblings; WT, wild-type)

three samples. Among, miRNAs -125b, -146a, -15a, and -342 in neocortex–hippocampus, -101a, -125b, -140, -146a, -15a, and -34a in eye tissues, and -140, -15a, -342, and -34a in tear fluids were significantly up-regulated. Similar to old Tg, miR-101a showed significant down-regulation in the neocortex–hippocampus of old NTg. As seen in the results from Tg mice (Table 3), NTg versus WT comparison revealed that miRNAs expression levels were generally similar across neocortex–hippocampus, eye tissues, and tear fluids at young and old ages.

### miRNAs relative expression levels in neocortex–hippocampus, eye tissues, and tear fluids with ageing and/or disease progression

#### Old versus young Tg mice
Over time, miRNAs relative expression levels were generally increased in old Tg mice (Table 5 and Fig 6 first column). In the neocortex–hippocampus, miRNAs -125b, -140, -15a, and -374c showed significant up-regulations, and miRNAs -101a, -146a, -342, and -34a, showed significant down-regulations. In the eye tissues,

all eight tested miRNAs showed significant or a trend toward up-regulation. Among, miRNAs -101a, -125b, -146a, -15a, -342, -34a, and -374c were significantly up-regulated. Likewise, all 8 tested miRNAs showed significant up-regulation in the tear fluids, suggesting their possible translational potential with disease progression as confirmed via cortical Aβ load along with reactive astrogliosis.

#### Old versus young NTg siblings
Over time, as like Tg mice, miRNAs relative expression levels were generally increased in old NTg siblings (Table 5 and Fig 6 second column). In the neocortex–hippocampus, miRNAs -125b, -146a, -15a, and -342 were significantly up-regulated and -101a was significantly down-regulated. In the eye tissues, miRNAs -101a, -125b, 140, -15a, and -34a showed significant up-regulations and in the tear fluids, excluding miR-374c, rest of the tested miRNAs showed significant up-regulations.

#### Old versus young WT controls
In contrast to Tg mice and NTg siblings, miRNAs relative expression levels were generally decreased in old WT controls (Table 5 and

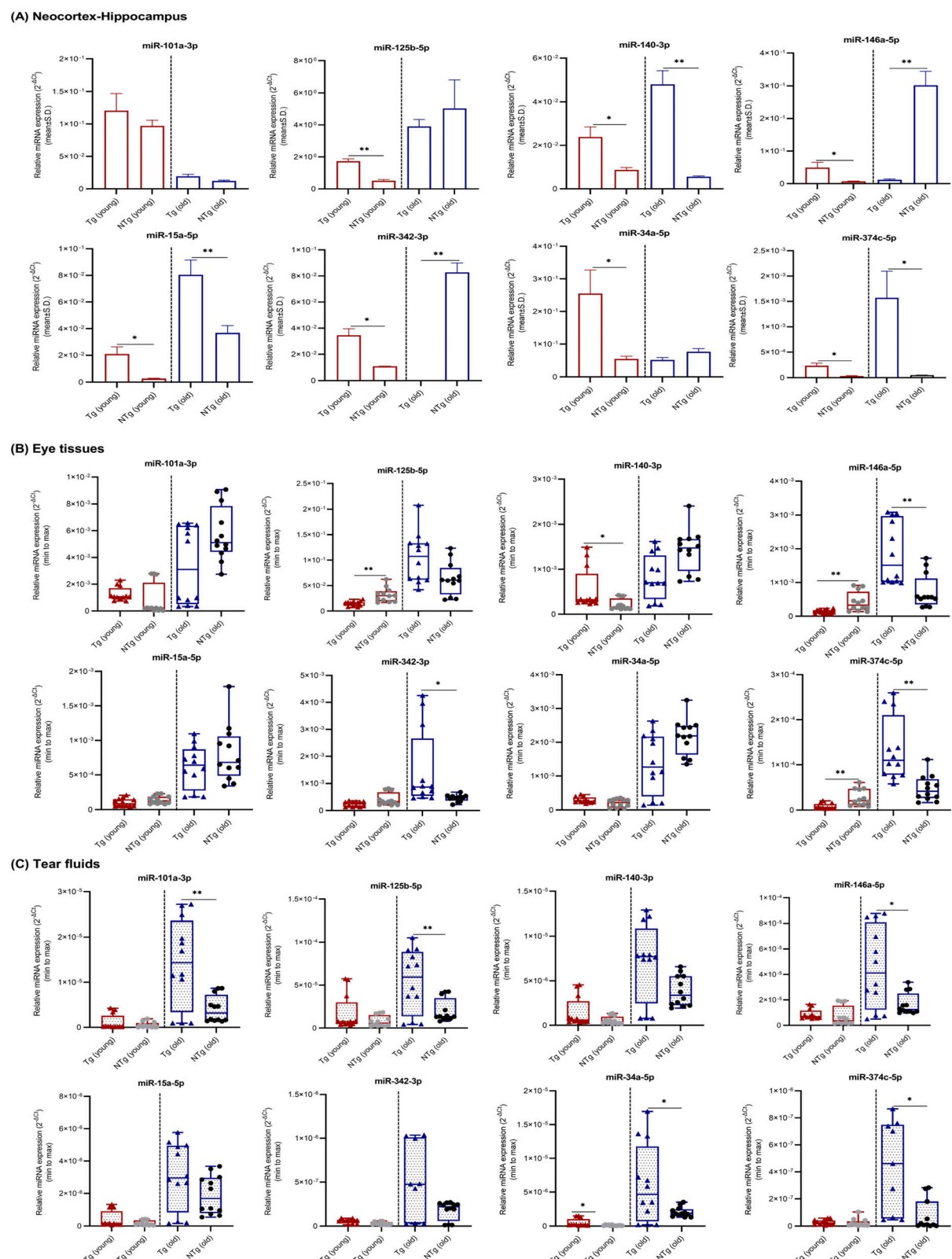

**Figure 3. miRNAs relative expression levels in neocortex–hippocampus, eye tissues, and tear fluid samples.**
**(A, B, C)** miRNAs relative expression levels (mean ± Std. $2^{-\Delta Ct}$) in (A) neocortex–hippocampus, (B) eye tissues, and (C) tear fluids of young (n = 4 per group, 12–16 wk, females) and old (n = 4 per group, 36–40 wk, females) Tg mice and NTg siblings are illustrated. Box plots (minimum to maximum) were used to illustrate relative expression levels in eye tissues and tear fluid samples. Individual data points were superimposed on each box plot. Differentially expressed miRNAs are indicated for each comparison based on two-fold intergroup difference (>2.0 or <0.5) and P-value as *P < 0.05, **P < 0.01 (Welch's t test).

**Table 2.** miRNA candidates relative expression levels in neocortex–hippocampus, eye tissues, and tear fluids of young (12–16 wk, n = 4 per group, females) and old (36–40 wk, n = 4 per group, females) Tg mice were compared with respective matched NTg siblings.

| miRNAs | Differentially expressed miRNAs (Log$_2$[FC] >1.0 or < −1.0 and $P$ < 0.05) | | | | | |
| --- | --- | --- | --- | --- | --- | --- |
| | Tg versus NTg (young) | | | Tg versus NTg (old) | | |
| | *N&H* | *Eye* | *Tear* | *N&H* | *Eye* | *Tear* |
| miR-101a | 0.31 (0.257) | 0.63 (0.254) | 1.08 (0.288) | 0.67 (0.038) | −0.77 (0.031) | **1.81 (0.004)** |
| miR-125b | **1.76 (0.001)** | **−1.10 (0.001)** | 1.09 (0.173) | −0.37 (0.385) | 0.71 (0.026) | **1.44 (0.009)** |
| miR-140 | **1.45 (0.024)** | **1.34 (0.034)** | 1.35 (0.101) | **3.10 (0.007)** | −0.79 (0.006) | 0.88 (0.029) |
| miR-146a | **2.79 (0.043)** | **−1.65 (0.005)** | 0.29 (0.53) | **−4.65 (0.007)** | **1.36 (0.001)** | **1.44 (0.011)** |
| miR-15a | **2.94 (0.028)** | −0.48 (0.099) | 1.13 (0.176) | **1.12 (0.009)** | −0.39 (0.216) | 0.68 (0.105) |
| miR-342 | **1.65 (0.014)** | −0.80 (0.027) | 0.91 (0.005) | **−11.08 (0.002)** | **1.70 (0.028)** | 1.49 (0.056) |
| miR-34a | **2.22 (0.039)** | 0.50 (0.053) | **2.14 (0.046)** | −0.57 (0.022) | −0.67 (0.015) | **1.56 (0.033)** |
| miR-374c | **2.81 (0.017)** | **−1.79 (0.004)** | 0.20 (0.732) | **4.91 (0.038)** | **1.49 (0.001)** | **2.51 (0.013)** |

Differentially expressed miRNAs are indicated (bold) based on $P$ < 0.05 (Welch's $t$ test, two-tailed) and log$_2$[FC] >1.0 considered up-regulated (black color) and log$_2$[FC] <−1.0 considered down-regulated (blue color). (N&H, neocortex–hippocampus; FC, fold change)

Fig 6 third column). Importantly, excluding miR-140, rest of the miRNAs were significantly down-regulated in the neocortex–hippocampus. Similarly, most of the tested miRNAs showed significant or a trend toward down-regulation in the eye tissues and tear fluids. miRNA -125b in eye tissues and miRNAs -15a, -342, -34a, and -374c in tear fluids were significantly down-regulated.

Because the AD-related neuropathological changes primarily affect neocortex–hippocampus in early stages, miRNAs relative expression levels obtained for other brain regions such as olfactory bulb, brainstem, and cerebellum were given under supplementary materials (Tables S1–S4 and Figs S1–Figs S4). Region 3 (striatum, thalamus, and hypothalamus) was excluded because of the possible cross contamination of other brain regions at dissection.

### TargetScan generated miRNA–mRNA interactions

Our TargetScan search generated several functionally related cluster miRNAs that shared conserved seed regions with AD-associated genes (Table 6 and Fig 7). In addition, some individual miRNAs are shown to target several AD-associated genes (Table 6 and Fig 7). We identified potential miRNA–mRNA interactions for our tested miRNAs including miR-101a and *APP* and *SORL1*; miR-125b and *BCL2*; miR-146a and *MAPT*, *CFH*, and *ATG12*; miR-15a and *APP*, *BACE1*, and *BCL2*; miR-34a and *MAPT* and *BCL2*; miR-342 and *BCL2*, *SORL1*, and *CACNA1C*; miR-140 and *BACE1*, *BCL2*, and *SIRT1*; and miR-374c and *APP*, *BACE1*, *PSEN1*, *CACNA1C*, *BCL2*, and *ATG12*.

## Discussion

Premortem diagnosis of AD relies on costly and/or highly invasive tests (e.g., PET, MRI, and CSF biomarkers) that are not appropriate for screening at a population level. Our main goal of the study is to determine the translational potential of tear fluid miRNAs as noninvasive early diagnostic biomarkers by using a transgenic AD mouse model. In the literatures, there exist inconsistencies with respect to deregulated miRNAs associated with AD, making it

difficult to use miRNAs as potential diagnostic and prognostic biomarkers. There are several reasons for such discrepancies: variations in types of biological samples used (e.g., different brain regions, CSF, serum, plasma, blood, and other body fluids); variations in sampling time points (e.g., early, mid, and later stages of the disease progression); variations in transgenic AD animal models and their controls (e.g., APP-PSEN1 mice, 5XFAD mice, 3XTG-AD mice, and C57BL/6J AD mice); variations in techniques used to determine the level of expression; and methods used to analyze the data could substantially affect the translational potential of these miRNAs as biomarkers. There are two different APP-PS1 double transgenic mice: B6;C3-Tg (strain#034829-JAX) and B6.Cg-Tg (strain#034832-JAX) that have been widely studied for miRNA investigations in AD. For B6;C3-Tg mice, its noncarrier siblings are recommended as suggested controls. For B6.Cg-Tg mice, its noncarrier siblings are suggested controls and C57BL/6J mice are approximate controls. However, in the published literatures, starin/control information is generally incomplete. Therefore, in this study, we systematically investigated the miRNAs expression levels in tissue and tear fluid samples using age- and sex-matched APP-PS1 mice (B6;C3-Tg), its noncarrier siblings and C57BL/6J mice, the most widely used background strain.

Our gene expression data showed a consistent pattern between neocortex–hippocampus and eye tissues for human and mouse *APP/App* and *PSEN1/Psen1* mRNAs at young and old ages (Table 1 and Fig 2). Furthermore, those mRNAs relative expression levels were comparable between NTg siblings and WT controls. Our gene data also supported the mutant human *PSEN1*-mediated amyloid hypothesis in the transgenic APP-PS1 mouse model. Hardy and Selkoe (2002)'s amyloid hypothesis proposed that *PSEN1* mutations initiate disease pathogenesis by increasing the production of Aβ42. PSEN1 functions as a catalytic subunit of γ-secretase, an intramembranous protease that cleaves APP. Although Aβ40 accounts for ~90% of Aβ production, the minor Aβ42 product is more hydrophobic and is thought to nucleate Aβ aggregation, leading to amyloid plaque deposition in the AD brain (Borchelt et al, 1996; Duff et al, 1996; Hardy & Selkoe, 2002).

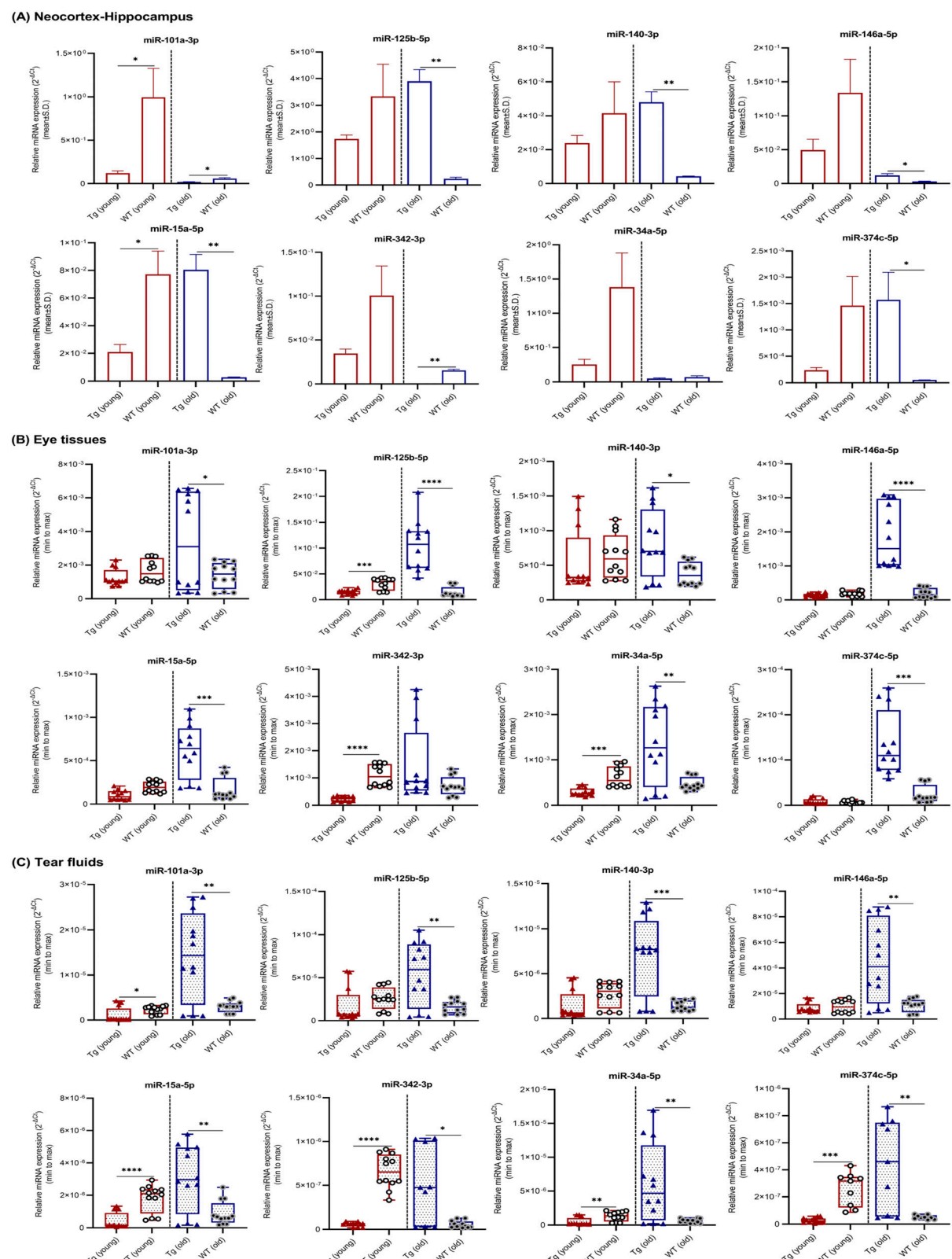

**Figure 4. miRNAs relative expression levels in neocortex–hippocampus, eye tissues, and tear fluid samples.**
**(A, B, C)** miRNAs relative expression levels (mean ± Std. $2^{-\Delta Ct}$) in (A) neocortex–hippocampus, (B) eye tissues, and (C) tear fluids of young (n = 4 per group, 12–16 wk, females) and old (n = 4 per group, 36–40 wk, females) Tg mice and C57BL/6J WT controls are illustrated. Box plots (minimum to maximum) were used to illustrate relative expression levels in eye tissues and tear fluid samples. Individual data points were superimposed on each box plot. Differentially expressed miRNAs are indicated for each comparison based on two-fold intergroup difference (>2.0 or <0.5) and P-value as *P < 0.05, **P < 0.01, ***P < 0.001, ****P < 0.0001 (Welch's t test).

**Table 3.** miRNA candidates relative expression levels in neocortex–hippocampus, eye tissues, and tear fluids of young (12–16 wk, n = 4 per group, females) and old (36–40 wk, n = 4 per group, females) Tg mice were compared with respective matched C57BL/6J WT controls.

| miRNAs | Differentially expressed miRNAs (Log$_2$[FC] >1.0 or < −1.0 and $P$ < 0.05) | | | | | |
|---|---|---|---|---|---|---|
| | Tg versus WT (young) | | | Tg versus WT (old) | | |
| | N&H | Eye | Tear | N&H | Eye | Tear |
| miR-101a | **−3.04 (0.044)** | −0.40 (0.118) | **−1.09 (0.038)** | **−1.59 (0.013)** | **1.27 (0.039)** | **2.25 (0.002)** |
| miR-125b | −0.94 (0.144) | **−1.05 (<0.001)** | −0.60 (0.225) | **4.04 (0.004)** | **2.84 (<0.001)** | **1.75 (0.004)** |
| miR-140 | −0.80 (0.235) | −0.20 (0.619) | −0.93 (0.05) | **3.48 (0.006)** | **1.12 (0.011)** | **2.32 (0.001)** |
| miR-146a | −1.43 (0.085) | −0.43 (0.097) | −0.17 (0.564) | **1.96 (0.022)** | **3.24 (<0.001)** | **2.05 (0.003)** |
| miR-15a | **−1.88 (0.02)** | −0.95 (0.001) | **−2.20 (<0.001)** | **4.85 (0.007)** | **1.90 (<0.001)** | **1.80 (0.003)** |
| miR-342 | −1.54 (0.074) | **−2.20 (<0.001)** | **−3.53 (<0.001)** | **−8.66 (0.003)** | 1.07 (0.088) | **3.15 (0.014)** |
| miR-34a | −2.44 (0.057) | **−1.09 (<0.001)** | **−1.44 (0.005)** | −0.47 (0.209) | **1.54 (0.005)** | **3.31 (0.007)** |
| miR-374c | −2.62 (0.061) | 0.18 (0.639) | **−3.26 (<0.001)** | **4.91 (0.038)** | **2.48 (<0.001)** | **3.10 (0.009)** |

Differentially expressed miRNAs are indicated (bold) based on $P$ < 0.05 (Welch's $t$ test, two-tailed) and log$_2$[FC] >1.0 considered up-regulated (black color) and log$_2$[FC] <−1.0 considered down-regulated (blue color). (N&H, neocortex–hippocampus, FC, fold change)

Of the 10 tested miRNAs, miRNAs -302c and -653 were undetermined in all three mice groups, suggesting their absence or low abundance nature in tested samples. Remaining eight miRNAs revealed a similar expression pattern for both Tg mice and NTg siblings when compared with WT controls. Although, Tg mice share similar genetic background with NTg siblings except two inserted human transgenes, the subsequent upstream or downstream effects of those two inserted human transgenes are not assured. Therefore, by comparing Tg mice versus NTg siblings, which could possibly have resulted from underlying molecular etiology of AD, the translational potential of tear fluid miRNAs based on its relative expression level, direction of regulation with disease progression, and their similarities to neocortex–hippocampus were explored.

On the whole, tear fluid miRNAs seem to be promising candidates for AD biomarker discovery. We notice a strong correlation in the expression levels of miRNAs between Tg mouse tear fluids and literature-based human CSF findings (Table S5). In early stage, all eight tested miRNAs were significantly down-regulated or showed a trend toward down-regulation in young Tg and NTg versus young WT (Tables 3 and 4). However, between young Tg and NTg, miR-34a showed a significant up-regulation and others showed a trend toward up-regulation in Tg mice. miR-34a is recognized as a proinflammatory miRNA involved in neurodegeneration (Lukiw et al, 2008; Zhao et al, 2015a; Pogue & Lukiw, 2018). miR-34a expression needs to be optimal in neurons, as an aberrant increase or decrease in its expression causes apoptosis (Modi et al, 2016). miR-34a had shown significant up-regulation in the CSF samples of AD patients (Lukiw et al, 2012). On the other hand, all the tested miRNAs were significantly up-regulated or showed a trend toward up-regulation in the tear fluids of old Tg versus old NTg and WT (Tables 2 and 3). Although old NTg siblings revealed a similar pattern when compared with old WT controls (Table 4), those miRNAs relative expression levels were considerably high in the tear fluids of old Tg mice (Table 2 and Fig 3C). Particularly, anti-amyloidogenic miR-101a (Vilardo et al, 2010; Long & Lahiri, 2011; Li et al, 2019), proinflammatory miRNAs -125b, -146a, and -34a (Lukiw et al, 2008; Zhao et al, 2015a), and the novel miR-374c were significantly

up-regulated. Significantly up-regulated miRNAs -146a (Lukiw et al, 2012; Alexandrov et al, 2012; Denk et al, 2015; Gong & Sun, 2022) and -125b (Alexandrov et al, 2012; Lukiw et al, 2012; Jin et al, 2018; Mckeever et al, 2018) have been reported in the CSF samples of AD patients. Furthermore, by comparing old versus young mice (Table 5 and Fig 6C), we suggest that secretion or circulating level of these miRNAs and their stability seem to be high in old Tg mice. In contrast, excluding miR-101a and -146a, other tested miRNAs were significantly down-regulated or showed a trend toward down-regulation in the tear fluids of old WT controls. C57BL/6J is a widely used mouse model for diet-induced obesity, type 2 diabetes, atherosclerosis, and age-related hearing loss. Therefore, our findings suggest that ageing does not hold a huge impact on tested miRNAs expression levels in tear fluids.

Similar to the findings obtained for tear fluids, in early stage, all the tested miRNAs showed significant or a trend toward down-regulation in the neocortex–hippocampus of young Tg and NTg versus young WT, and those levels were mostly reversed over time (Tables 3 and 4). In contrast, all miRNAs except miR-101a showed significant up-regulation in young Tg when compared with young NTg (Table 2 and Fig 3A). Based on this observation, we propose that disease-specific miRNAs' deregulations or underlying molecular changes occur at an early stage (3–4 mo) before the development of pathological lesions. miRNAs -140, -15a, and -374c showed significant up-regulations in the neocortex–hippocampus of young and old Tg mice when compared with matched NTg siblings (Table 2) and with disease progression (Table 5). Based on their common target genes such as *BACE1* and *BCL2* identified via TargetScan search (Table 6 and Fig 7) and from the literature (Liu et al, 2019; Zhang et al, 2020), these miRNAs appear to maintain a balance between neuroprotection and neurodegeneration. Other than that, miRNAs -125b, -146a, -34a, and -342 also showed significant up-regulations in the neocortex–hippocampus of young Tg mice. There have been studies that reported significantly up-regulated miR-146a in human AD-affected temporal lobe (Wang et al, 2016; Pogue & Lukiw, 2018) and hippocampus (Jaber et al, 2017), miR-125b in human AD-affected temporal lobe (Brodmann area 22) (Pogue &

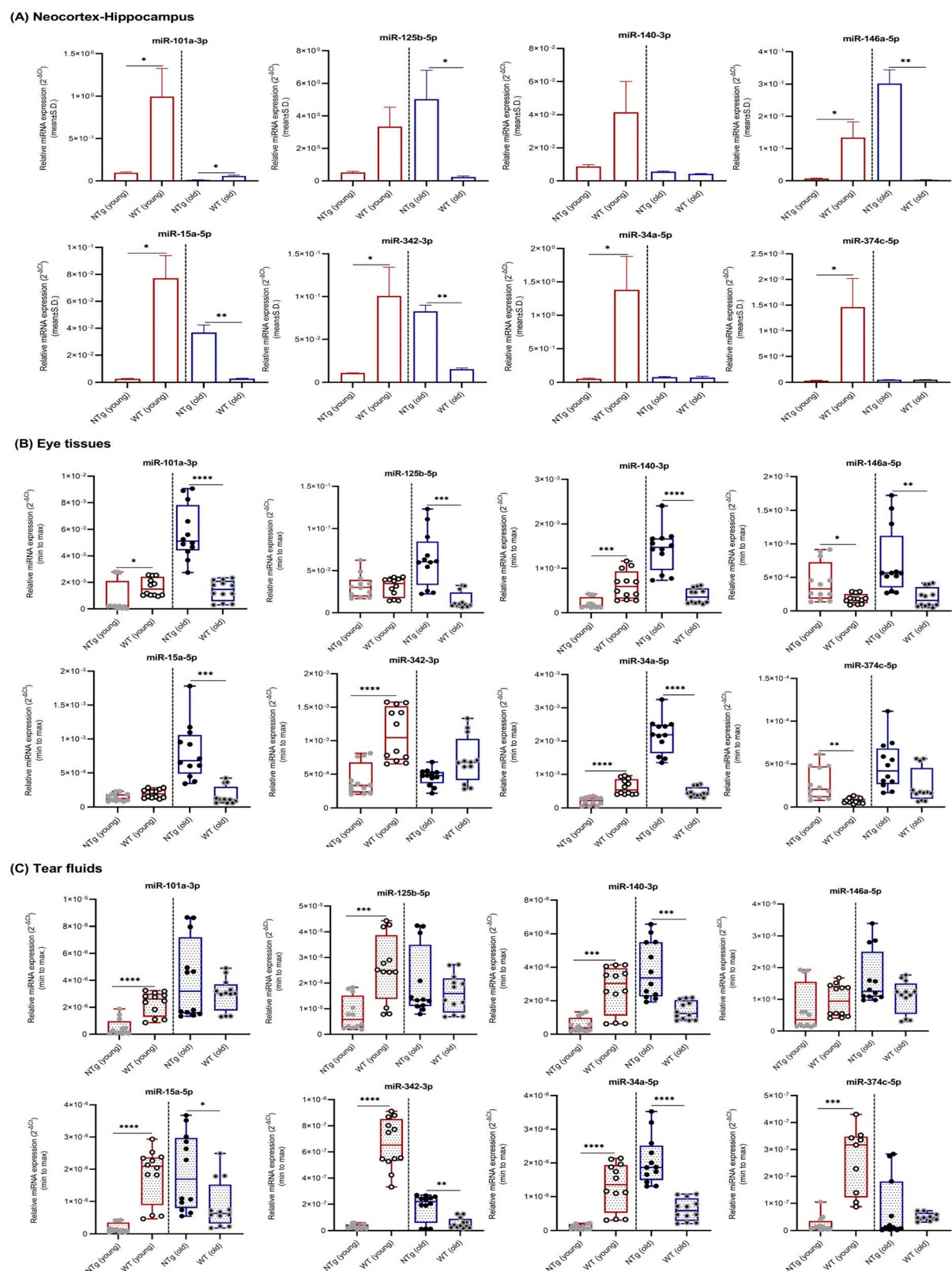

**Figure 5. miRNAs relative expression levels in neocortex–hippocampus, eye tissues, and tear fluid samples.**
**(A, B, C)** miRNAs relative expression levels (mean ± Std. $2^{-\Delta Ct}$) in (A) neocortex–hippocampus, (B) eye tissues, and (C) tear fluids of young (n = 4 per group, 12–16 wk, females) and old (n = 4 per group, 36–40 wk, females) NTg siblings and C57BL/6J WT controls are illustrated. Box plots (minimum to maximum) were used to illustrate relative expression levels in eye tissues and tear fluid samples. Individual data points were superimposed on each box plot. Differentially expressed miRNAs are indicated for each comparison based on two-fold intergroup difference (>2.0 or <0.5) and P-value as *P < 0.05, **P < 0.01, ***P < 0.001, ****P < 0.0001 (Welch's t test).

**Table 4.** miRNA candidates relative expression levels in neocortex–hippocampus, eye tissues, and tear fluids of young (12–16 wk, n = 4 per group, females) and old (36–40 wk, n = 4 per group, females) NTg siblings were compared with respective matched C57BL/6J WT controls.

| miRNAs | Differentially expressed miRNAs (Log2[FC] >1.0 or < −1.0 and P < 0.05) | | | | | | |
| --- | --- | --- | --- | --- | --- | --- | --- |
| | NTg versus WT (young) | | | NTg versus WT (old) | | | |
| | N&H | Eye | Tear | N&H | Eye | Tear | |
| miR-101a | **−3.36 (0.042)** | **−1.03 (0.047)** | **−2.17 (<0.001)** | **−2.26 (0.013)** | **2.04 (<0.001)** | 0.44 (0.259) | |
| miR-125b | −2.70 (0.054) | 0.05 (0.845) | **−1.69 (<0.001)** | **4.41 (0.042)** | **2.12 (<0.001)** | 0.30 (0.401) | |
| miR-140 | −2.25 (0.091) | **−1.54 (0.001)** | **−2.28 (<0.001)** | 0.37 (0.017) | **1.91 (<0.001)** | **1.44 (<0.001)** | |
| miR-146a | **−4.23 (0.046)** | **1.23 (0.015)** | −0.46 (0.32) | **6.61 (0.006)** | **1.88 (0.004)** | 0.61 (0.062) | |
| miR-15a | **−4.82 (0.016)** | −0.47 (0.036) | **−3.33 (<0.001)** | **3.72 (0.008)** | **2.28 (<0.001)** | **1.12 (0.019)** | |
| miR-342 | **−3.19 (0.044)** | **−1.40 (<0.001)** | **−4.45 (<0.001)** | **2.42 (0.003)** | −0.63 (0.031) | **1.66 (0.002)** | |
| miR-34a | **−4.66 (0.044)** | **−1.59 (<0.001)** | **−3.57 (<0.001)** | 0.11 (0.685) | **2.20 (<0.001)** | **1.74 (<0.001)** | |
| miR-374c | **−5.43 (0.047)** | **1.98 (0.003)** | **−3.45 (<0.001)** | 0.00 (0.995) | 0.99 (0.026) | 0.60 (0.461) | |

Differentially expressed miRNAs are indicated (bold) based on $P < 0.05$ (Welch's $t$ test, two-tailed) and $\log_2$[FC] >1.0 considered up-regulated (black color) and $\log_2$[FC] <−1.0 considered down-regulated (blue color). (N&H, neocortex–hippocampus, FC, fold change)

**Table 5.** miRNA candidates relative expression levels in neocortex–hippocampus, eye tissues, and tear fluids with ageing and/or disease progression old (36–40 wk, n = 4 per group, females) Tg mice, NTg siblings, and C57BL/6J WT controls were compared with matched young (12–16 wk, n = 4 per group, females) Tg mice, NTg siblings, and WT controls, respectively.

| miRNAs | Differentially expressed miRNAs (Log2[FC] >1.0 or < −1.0 and P < 0.05) | | | | | | | | |
| --- | --- | --- | --- | --- | --- | --- | --- | --- | --- |
| | Old versus young Tg | | | Old versus young NTg | | | Old versus young WT | | |
| | N&H | Eye | Tear | N&H | Eye | Tear | N&H | Eye | Tear |
| miR-101a | **−2.64 (0.021)** | **1.43 (0.028)** | **3.72 (0.001)** | **−3.00 (0.003)** | **2.83 (<0.001)** | **2.99 (0.002)** | **−4.10 (0.039)** | −0.24 (0.398) | 0.37 (0.124) |
| miR-125b | **1.17 (0.008)** | **2.82 (<0.001)** | **1.67 (0.007)** | **3.29 (0.047)** | **1.01 (0.006)** | **1.32 (0.013)** | **−3.82 (0.046)** | **−1.07 (0.001)** | −0.68 (0.036) |
| miR-140 | **1.01 (0.007)** | 0.60 (0.167) | **2.32 (0.001)** | −0.64 (0.026) | **2.73 (<0.001)** | **2.79 (<0.001)** | −3.27 (0.073) | −0.72 (0.03) | −0.93 (0.009) |
| miR-146a | −2.04 (0.051) | **3.78 (<0.001)** | **2.39 (0.002)** | **5.40 (0.007)** | 0.76 (0.091) | **1.24 (0.008)** | **−5.44 (0.044)** | 0.11 (0.745) | 0.17 (0.57) |
| miR-15a | **1.94 (0.004)** | **2.60 (<0.001)** | **2.96 (0.001)** | **3.75 (0.008)** | **2.50 (<0.001)** | **3.41 (<0.001)** | **−4.79 (0.016)** | −0.25 (0.468) | **−1.04 (0.008)** |
| miR-342 | **−9.82 (0.007)** | **2.65 (0.011)** | **3.14 (0.015)** | **2.91 (0.003)** | 0.15 (0.569) | **2.57 (<0.001)** | **−2.70 (0.048)** | −0.62 (0.019) | **−3.54 (<0.001)** |
| miR-34a | **−2.31 (0.038)** | **2.22 (0.001)** | **3.66 (0.006)** | 0.49 (0.037) | **3.39 (<0.001)** | **4.23 (<0.001)** | **−4.28 (0.045)** | −0.40 (0.063) | **−1.09 (0.007)** |
| miR-374c | **2.72 (0.047)** | **4.09 (<0.001)** | **4.01 (0.006)** | 0.62 (0.067) | 0.80 (0.048) | 1.71 (0.152) | **−4.81 (0.048)** | **1.79 (0.01)** | **−2.34 (0.001)** |

Differentially expressed miRNAs are indicated (bold) based on $P < 0.05$ (Welch's t test, two-tailed) and $\log_2$[FC] >1.0 considered up-regulated (black color) and $\log_2$[FC] <−1.0 considered down-regulated (blue color). (N&H, neocortex–hippocampus; FC, fold change)

Lukiw, 2018) and frontal cortex (Brodmann areas 6 and 8) (Banzhaf-Strathmann et al, 2014), miR-34a in human AD-affected temporal lobe (Sarkar et al, 2016; Jaber et al, 2017; Pogue & Lukiw, 2018), and miR-342 in the whole brain of APP-PS1 mice (Wang et al, 2017). miR-146a is implicated in inflammatory neurodegeneration (Lukiw et al, 2008; Zhao et al, 2015a). Wang et al's (2016) in vitro and in vivo studies demonstrated that miR-146a contributes to $\tau$ hyper-phosphorylation and AD pathogenesis via inhibition of rho-associated, coiled-coil containing protein kinase 1 (ROCK1). Gain-of-function studies for miR-146a suggest its crucial role in controlling genetic pathways essential to innate immune responses, inflammation, and microglial activation state, which are major features in the pathogenesis of AD (Fan et al, 2020). Similarly, mRNA targets of miR-125b are involved in pathogenic immune and inflammatory signaling in the brain and retina (Zhao et al, 2015a; Pogue & Lukiw, 2018). miR-34a targets triggering receptor expressed in myeloid/microglial cells-2 which is a critical component in A$\beta$42-peptide

clearance (Bhattacharjee et al, 2016). miR-34a mediated amyloi-dogenic processing of APP (Jian et al, 2017) and its inhibitory role on translation of anti-apoptosis regulating protein Bcl-2 leading to neurodegeneration (Wang et al, 2009) have been reported. Therefore, our findings suggest that deregulated miRNAs -125b, -146a, -342, and -34a in neocortex–hippocampus may relate with a pathogenic inflammatory immune response in young Tg mice.

Young Tg versus NTg comparison showed significant up-regulations for all the tested miRNAs except miR-101a (Table 2). miR-101a is one of the brain-enriched miRNAs (Shao et al, 2010). miR-101a showed significant down-regulation in the neocortex–hippocampus of old Tg and NTg when compared with old WT. Although miR-101a did not show a significant difference in the neocortex–hippocampus of old Tg mice when compared with old NTg siblings, it showed significant down-regulations in other brain regions such as olfactory bulb, brainstem, and cerebellum (Table S2). Furthermore, regardless of mice groups, miR-101a

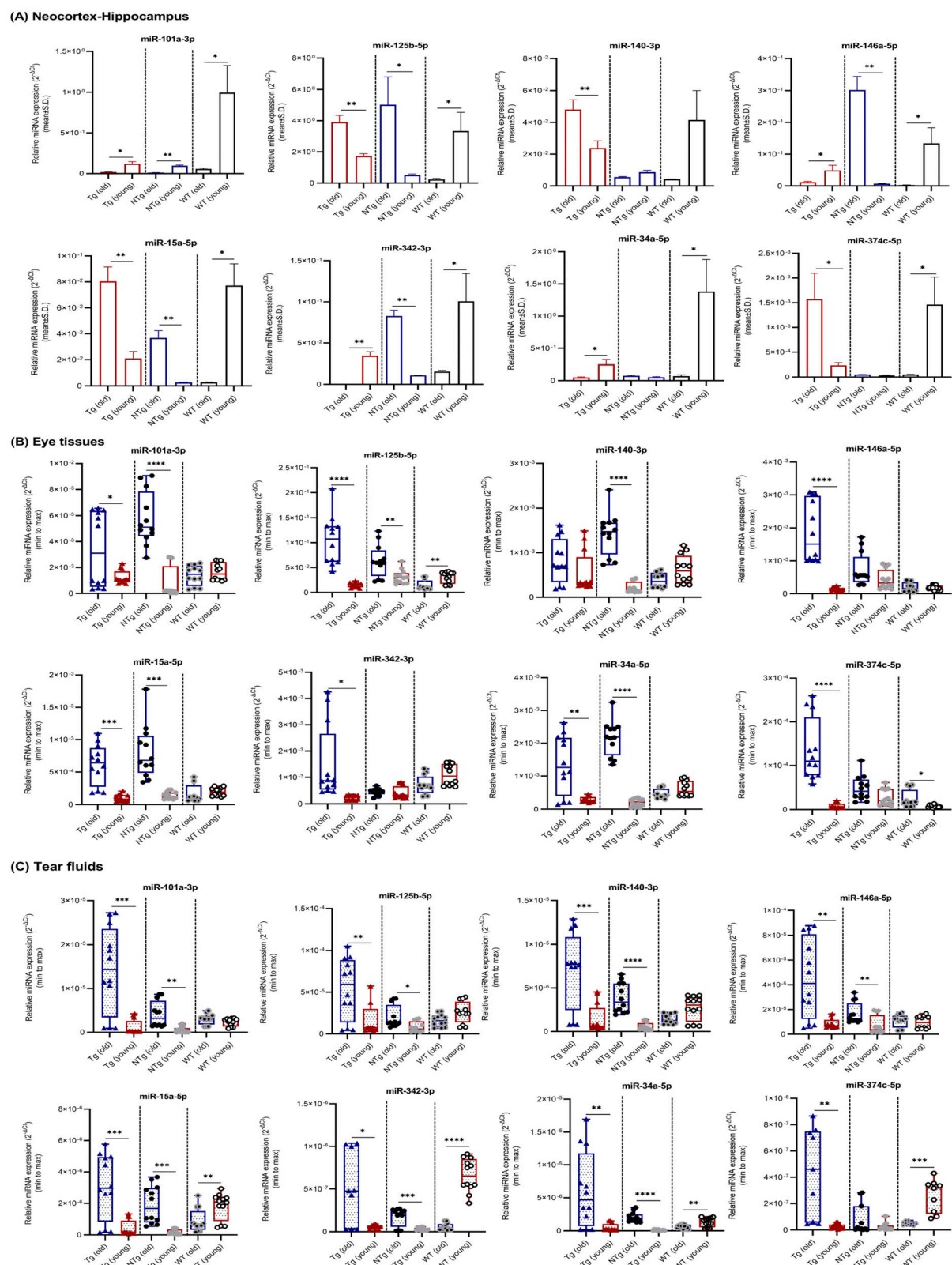

**Figure 6. miRNAs relative expression levels in neocortex–hippocampus, eye tissues, and tear fluids with ageing and/or disease progression.**
**(A, B, C)** miRNAs relative expression levels (mean ± Std. $2^{-\Delta Ct}$) in (A) neocortex–hippocampus, (B) eye tissues, and (C) tear fluids of old (n = 4 per group, 36–40 wk, females) and young (n = 4 per group, 12–16 wk, females) Tg mice, NTg siblings, and C57BL/6J WT controls are illustrated. Box plots (minimum to maximum) were used to illustrate relative expression levels in eye tissues and tear fluid samples. Individual data points were superimposed on each box plot. Differentially expressed miRNAs are indicated for each comparison based on two-fold intergroup difference (>2.0 or <0.5) and *P*-value as \**P* < 0.05, \*\**P* < 0.01, \*\*\**P* < 0.001, \*\*\*\**P* < 0.0001 (Welch's *t* test).

**Table 6.  TargetScan-generated miRNA–mRNA interactions.**

| Functionally related cluster miRNAs | Target mRNAs (>1) | Single miRNA | Target mRNAs (>1) |
|---|---|---|---|
| -302a/b/c/d/e, -520a/b/c/d/e/f, -372-3p, and -373-3p | *APP* and *PSEN1* | -532-3p | *APP, PSEN1, SORL1,* and *MAPT* |
| -15a/b, -16-5p, -195-5p, -424-5p, -497-5p, -4770, and -6838-5p | *APP, BACE1,* and *BCL2* | -140-3p | *BACE1, BCL2,* and *SIRT1* |
| -17-5p, -20a/b, -93-5p, -106a/b, -185-5p, -519d-3p, -526b-3p, -873-5p, -4306, and -4644 | *APP, PSEN1,* and *SORL1* | -448 | *BCL2, CACNA1C, SORL1,* and *SIRT1* |
| -let-7a/b/f, -26a/b, -98-3p, -582-5p, -1297, -4465, and -6835-3p | *APP* and *CACNA1C* | -330-3p | *PSEN1, PSEN2,* and *SORL1* |
| -96-5p, 200b/c, -365a/b, -429, -382-3p, and -1271-5p | *CACNA1C* and *BCL2* | -9-5p | *BACE1, PSEN1,* and *SIRT1* |
| -25-3p, -32-5p, -92a/b, -140-5p, -363-3p, and -367-3p | *SORL1* and *CACNA1C* | -342-3p | *BCL2, CACNA1C,* and *SORL1* |
| -124-3p.2, -141-3p, -200a-3p, and -506-3p | *BACE1* and *SIRT1* | -494-3p | *CACNA1C, SIRT1,* and *SORL1* |
| -181a/b/c/d and -4262 | *MAPT, SIRT1,* and *BCL2* | -323a-3p | *APP, BCL2,* and *CACNA1C* |
| -30a/b/c/d/e | *PSEN2, BCL2, SIRT1,* and *CACNA1C* | -488-3p | *PSEN1* and *SORL1* |
| -34a/b and -449a/b | *BCL2* and *MAPT* | -489-3p | *SORL1* and *MAPT* |
| -128-3p, -216a-3p, and -3681-3p | *PSEN1, SORL1,* and *SIRT1* | -137 | *CACNA1C* and *SORL1* |
| -103a-3p, -107 and -377-3p | *PSEN1* and *CACNA1C* | -505-3p.2 | *CACNA1C* and *ATG12* |
| -153-3p, -300 and -381-3p | *APP, CACNA1C,* and *SORL1* | -183-5p.2 | *CACNA1C* and *PSEN2* |
| -369-3p and -876-5p | *CACNA1C* and *BACE1* | -760 | *CACNA1C* and *MAPT* |
| -124-3p.1 and -135a/b | *BACE1, SORL1,* and *SIRT1* | -3167 | *CACNA1C* and *BACE1* |
| -29a/b/c | *BACE1, PSEN1,* and *SIRT1* | | |
| -146a/b and -7153-5p | *MAPT, CFH,* and *ATG12* | | |
| -485-5p and -6884-5p | *APOE, BACE1,* and *MAPT* | | |
| -132-3p and -212-3p | *MAPT* and *SIRT1* | | |
| -204-5p and -211-5p | *MAPT, SIRT1, CACNA1C,* and *BCL2* | | |
| -155-5p and -653-5p | *CACNA1C* and *SIRT1* | | |
| -202-5p and -383-5p | *BCL2* and *APP* | | |
| -374c-5p and -655-5p | *APP, BACE1, PSEN1, CACNA1C, BCL2,* and *ATG12* | | |
| -7-5p and -299-5p | *PSEN1* and *BACE1* | | |
| -143-3p and -6088 | *CACNA1C, BCL2,* and *BACE1* | | |
| -129-1-3p/-129-2-3p | *MAPT, BACE1,* and *SORL1* | | |
| -199a/b | *SIRT1* and *SORL1* | | |
| -19a/b | *CACNA1C, SORL1,* and *BACE1* | | |
| -27a/b | *BACE1, MAPT, PSEN1,* and *SORL1* | | |
| -101-3p.1/-101-3p.2 | *APP* and *SORL1* | | |

Functionally related cluster miRNAs that shared conserved seed regions with more than one target mRNAs and individual miRNA that shared conserved seed regions with more than one target mRNAs are summarized. *APP,* amyloid precursor protein; *PSEN1,* presenilin 1; *PSEN2,* presenilin 2; *BACE1,* beta-secretase 1; *BCL2,* B-cell lymphoma 2; *SORL1,* sortilin-related receptor 1; *MAPT,* microtubule-associated protein tau; *SIRT1,* sirtuin 1; *CACNA1C,* calcium voltage-gated channel subunit alpha1 C; *CFH,* complement factor H; *ATG12,* autophagy related 12; *APOE,* apolipoprotein E)

showed significant down-regulations in the neocortex–hippocampus of old mice when compared with their respective young ones (Table 5). Our TargetScan search revealed that miR-101a shares conserved seed regions with *APP* and *SORL1* (Table 6 and Fig 7). *SORL1* encodes an endocytic receptor involved in APP trafficking and processing (Rogaeva et al, 2007; DeRosa et al, 2022). In total, our findings suggest that excessive down-regulation of miR-101a in the neocortex–hippocampus and other brain regions may contribute pathological Aβ production in old Tg mice.

With disease progression as demonstrated via cortical Aβ load and reactive astrogliosis (Fig 1B), miR-125b showed consistent significant up-regulation, whereas -146a, -34a, and -342 showed significant down-regulations in the neocortex–hippocampus of old Tg mice (Table 5). In contrast, miRNAs -125b, -146a, -342, and -34a showed significant up-regulations or a trend toward up-regulation

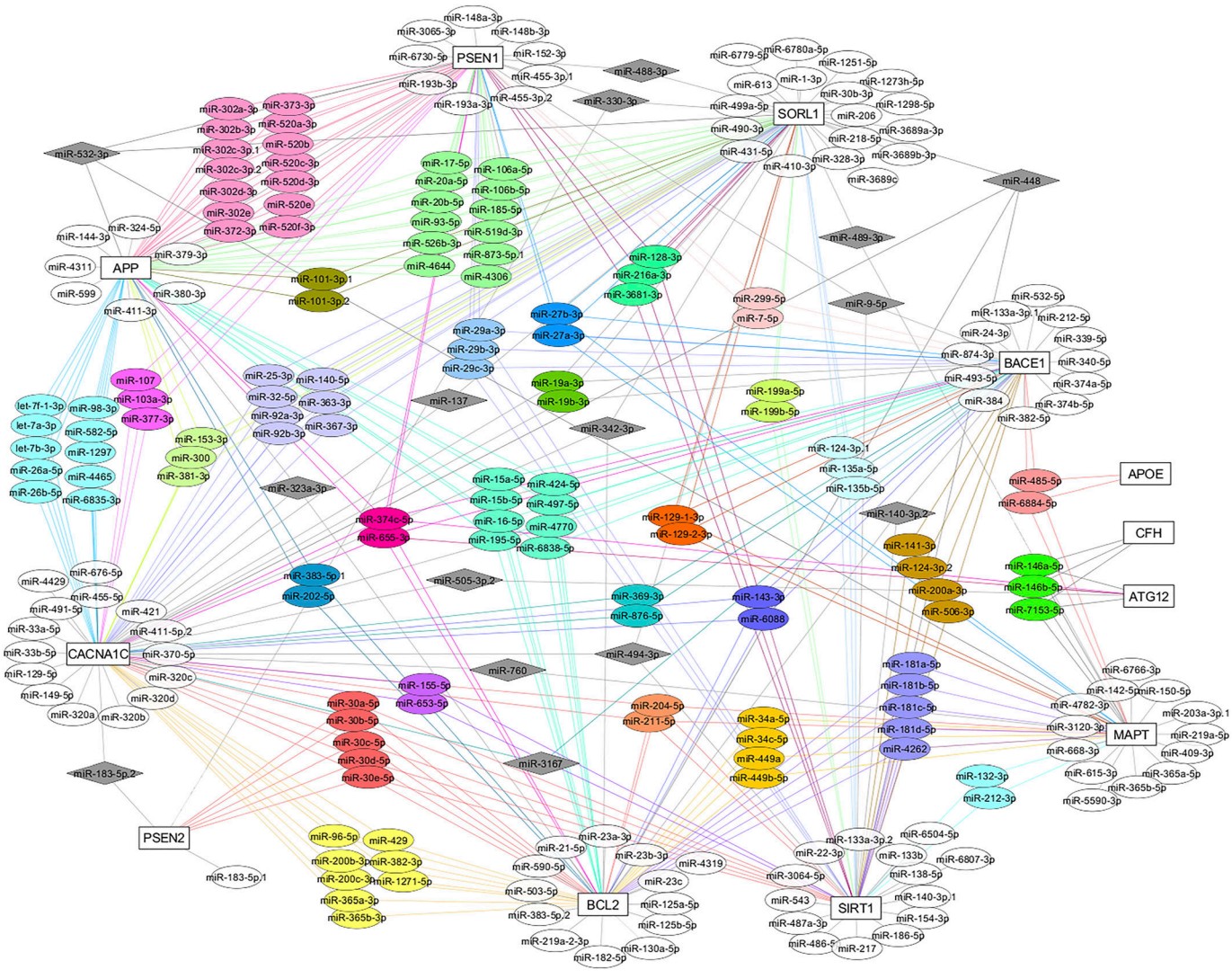

**Figure 7. TargetScan-predicted mRNA–miRNA interactions.**
The mRNA–miRNA interactions are illustrated using Cytoscape version 3.7.1 with manual adjustments. miRNAs that shared poorly conserved sequences were excluded. Source nodes are genes (square shape, white color) and the target nodes are miRNAs (ellipse shape). Functionally related cluster miRNAs (colored, ellipse shape) and individual miRNAs (grey, diamond shape) that shared conserved seed regions (8mer and 7mer) with more than one target genes and several miRNAs (white, ellipse shape) that target a single gene are illustrated.

in old NTg siblings (Table 5) where Aβ depositions were minimal or absent (Fig 1D). Our TargetScan search identified mRNA targets for these miRNAs based on their conserved seed regions (Table 6 and Fig 7). Both miRNAs -146a and -34a targeted *MAPT*. Dickson et al (2013) showed that miR-34a represses the expression of endogenous τ protein by binding 3′ UTR of human τ mRNA in a neuroblastoma cell line M17D. Similarly, miRNAs -125b, -342, and -34a targeted *BCL2*, an anti-apoptotic gene. Thus, they may involve in cellular and tissue level senescence and exhibit brain ageing and ageing phenotypes (Chen et al, 2021; Chua and Tang, 2019). However, other potential mRNA targets of these miRNAs such as *SORL1* and *MAPT* may play a crucial role in Tg mice leading to severe AD pathologies.

Of the eight expressed miRNAs, -140-3p and -374c-5p were novel candidates and both showed significant or a trend toward up-

regulation in tear fluids and neocortex–hippocampus of old Tg mice compared with old NTg siblings or old WT controls. Our TargetScan search identified several AD-associated mRNA targets for miR-374c (Table 6 and Fig 7). Other than that, organic cation transporter novel-type 1 (*OCTN1*) is a biological target of miR-374c (not included in Fig 7). Ergothioneine is a naturally occurring dietary antioxidant and its uptake and accumulation are purely dependent on OCTN1 distribution. Beneficial effects of ergothioneine uptake against Aβ accumulations have been reported in our previous studies (Whitmore et al, 2022; Wijesinghe et al, 2023). We also reported significantly depleted OCTN1 transporters in the whole eye of transgenic 5XFAD mice by comparing it with age-matched C57BL/6J WT controls (Wijesinghe et al, 2023). However, additional work is required to determine the functional relationship between miR-374c and OCTN1 in this context.

As a part of the CNS, the retina has been described as a window to the brain and intensively investigated to serve as a marker for AD. There are some functional similarities between tear fluids and CSF. Tear fluids are clear, colorless fluid covering the surface of the eye and provide protection, nourishment, waste removal and lubricant (Pflugfelder & Stern, 2020). Similarly, CSF is a clear colorless body fluid found within the tissue that surrounds brain and spinal cord and provides protection, nourishment, and waste removal. To our knowledge, this is the first comprehensive study that explored the translational potential of deregulated tear fluid miRNAs using a transgenic AD mouse model. Our data suggest strong correlations between the expression level of miRNAs in human CSF reported in the literatures and those found here in mouse tear fluids. Taken together, these data support the hypothesis that miRNA expression level in tear fluids may be useful as clinically relevant, noninvasive diagnostic, and prognostic biomarkers for AD.

## Materials and Methods

All the experiments were conducted in accordance with and approval of the University of British Columbia Animal Care Committee (A20-0150) and Biosafety Committee (B20-0074) recommendations.

APP/PS1 double transgenic (Tg) mice (B6;C3-Tg, strain#034829-JAX) expressing a chimeric mouse/human amyloid precursor protein (Mo/HuAPP695swe) and a mutant human presenilin 1 (PSEN1-dE9), its noncarrier siblings (NTg), and its background strain C57BL/6J WT controls were studied. Both mutations are associated with early-onset AD and amyloid plaque formation. The brain, whole eye, and tear fluid samples were collected from above three mice groups at two different ages: 12–16 wk (young) and 36–40 wk (old). Because the study was novel and intended to test the hypothesis, a total of 24 female mice were used in six groups: young Tg, old Tg, young NTg, old NTg, young WT, and old WT (n = 4 mice for each group) using resource equation method (Charan & Kantharia, 2013).

To test the study hypothesis across the brain, eye, and tear fluid samples, 10 miRNA candidates were selected including six already reported miRNAs -101a-3p, -125b-5p, 146a-5p, -15a-5p, -34a-5p, and -342-3p and four novel candidates -140-3p, -302c-3p, -374c-5p, and -653-5p at the time of study conceptualization (Table S6). Our selection was primarily based on the relevant literature and TargetScan search. Anti-amyloidogenic miR-101a (Vilardo et al, 2010; Long & Lahiri, 2011; Li et al, 2019) and -15a, which family miRNAs are anti-amyloidogenic (Liu et al, 2019; Zhang et al, 2020) were selected. In addition, three mostly reported proinflammatory miRNAs -125b, -146a, and -34a (Zhao et al, 2015a; Bhattacharjee et al, 2016; Pogue & Lukiw, 2018) were included. Above five miRNAs have been reported to be up-regulated or a trend toward up-regulation in the CSF samples of AD patients (Alexandrov et al, 2012; Lukiw et al, 2012; Denk et al, 2015; Sørensen et al, 2016; Jin et al, 2018; McKeever et al, 2018; Gong & Sun, 2022). Furthermore, miR-342-3p which showed consistent up-regulation in the whole brain samples of APP-PS1 mice at 1-, 3-, 6-, and 9- mo (Wang et al, 2017) was selected. For TargetScanMouse 7.2 (Agarwal et al, 2015) search, human genes associated with AD were used to identify their biological target miRNAs by searching for the presence of conserved sequences (8mer and 7mer) that match the seed region of each miRNA. miRNAs that shared poorly conserved sequences were excluded. Based on TargetScan search, miRNAs -140-3p which is abundant in the brain (Shao et al, 2010) and shared conserved seed regions with BACE1, BCL2, and SIRT1; -302c-3p which is one of the representative large functionally related cluster miRNAs -302a/b/c/d/e, -520a/b/c/d/e/f, -372-3p, and -373-3p those shared conserved seed regions with APP and PSEN1; -374c-5p which shared conserved seed regions with many AD-associated genes including APP, BACE1, PSEN1, CACNA1C, BCL2, and ATG12 and -653 which is functionally related to another proinflammatory miR-155-5p (Zhao et al, 2015a; Pogue & Lukiw, 2018) based on its conserved seed regions with CACNA1C and SIRT1 were selected. The mRNA–miRNA interactions were illustrated using Cytoscape version 3.7.1 (Shannon et al, 2003). miRNA–target mRNA interactions were manually adjusted to show the functionally related cluster miRNAs that shared conserved sequences with more than one target mRNAs.

### Tear fluids collection

We used a minimally invasive ethical approach to obtain the tear fluids from mice as described by Balafas et al (2019), with slight modifications. A subcutaneous injection of a mixture of anesthetic agents, ketamine hydrocholoride (80 mg/kg) and xylazine hydrocholoride (10 mg/kg) was given before tear fluids collection from each animal. Tear fluids were collected using sterile Schirmer tear test strips. Briefly, Schirmer tear test strip was cut by using a revolving leather hole punch device to achieve a circular shape, at a diameter of 2 mm. The circular shape was used so as to be atraumatic because of the smooth, round edges, avoiding damaging the ocular area (Balafas et al, 2019). The resulting discs were secured with mosquito forceps, and the smooth edge of the disc was placed inside the lower lid margin (inferior fornix) of the eye to collect tear gland secretion. Both eyes were used for tear fluids collection, one disc per eye. Once the discs were soaked fully in tears, they were immediately placed in Eppendorf tubes containing ~200 μl of water with RNase inhibitors (diethyl pyrocarbonate treated water), thoroughly mixed by hand for 30 s, transported on dry ice, and then stored at −80°C until use (Kenny et al, 2019).

### Brain and eye sample collection

Immediately after tear fluids collection, animals were euthanized, and the whole brain and both eye globes were quickly harvested and placed in 5 volumes of RNAlater (AM7021; Invitrogen). Next, the samples were dissected under a stereomicroscope (SMZ745T; Nikon) while keeping them in RNAlater. The left hemisphere of each mouse brain was dissected into five subregions, trying to match Thal's Aβ phases for senile plaques (Thal et al, 2002). This includes region 1—neocortex with hippocampus, region 2—olfactory bulb, region 3—striatum, thalamus, and hypothalamus, region 4—brainstem including midbrain, pons, and medulla, and region 5—cerebellum (Fig 8). Dissected mouse brains and the left eye globe were incubated overnight at 2–4°C in 5 volumes of fresh RNAlater. On the next day,

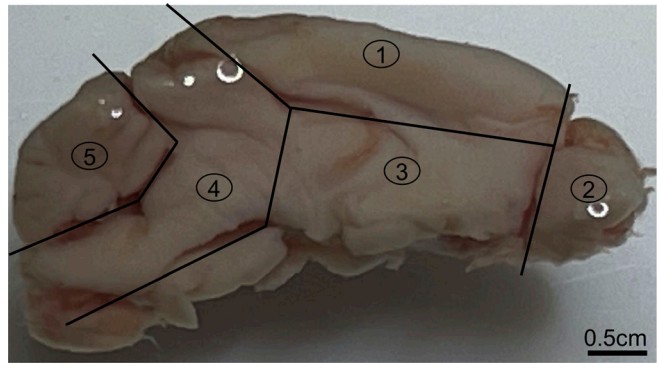

**Figure 8. Mouse brain mid-sagittal view.**
The mouse brain was dissected into five subregions. This includes region (1) neocortex with hippocampus; region (2) olfactory bulb; region (3) striatum, thalamus, and hypothalamus, region; (4) brainstem including midbrain, pons, and medulla; and region (5) cerebellum. Scale bar: 0.5 cm.

the supernatant was discarded, and the samples were transferred to –80°C until use. The right hemisphere of the mouse brain and right eye globe were preserved in 10% neutral buffered formalin for subsequent or future immunohistochemical studies.

### miRNA extraction from tear fluids and tissue samples

miRNeasy Serum/Plasma Kit (Cat# 217184; QIAGEN) was used to extract the miRNAs from tear fluids with slight modifications to manufacture's protocol. Briefly, samples were thawed on ice and vortexed thoroughly for 2–3 min. 200 $\mu$l of supernatant was used as the starting material, and 5 volumes QIAzol lysis reagent containing 1.25 $\mu$l bacteriophage MS2 RNA (Cat# 10165948001 Roche; Sigma-Aldrich) as per manufacturer's recommendation for fluids was added to each sample before homogenization. Content was vortexed for 1–2 min, and the homogenate was incubated at room temperature for 5 min, followed by internal control, 3.5 $\mu$l of 10 nM Cel-miR-39 RNA oligos (Cat# 10620310; Invitrogen) with 5' phosphate modification was added to each sample. Thereafter, an equal volume of chloroform (200 $\mu$l) was added, vortexed for 1 min, and incubated at room temperature for 2–3 min. Remaining steps were performed in accordance with manufacturer's instructions. miRNA extraction was individually carried out for each mouse's tear fluids.

miRNeasy mini kit (Cat# 217004; QIAGEN) was used to extract the miRNAs from brain and eye tissues as per the manufacturer's instructions. Half of the tissue homogenates were stored at –80°C for subsequent gene expression studies. miRNA extraction was individually carried out for each mouse's eye as like tear fluids. On the other hand, dissected brain tissues of same anatomical region were pooled based on age and mice group, and the miRNAs were extracted from pooled homogenized samples. It has been said that pooling of tissue samples from the same anatomical region of the brain and matched for mouse type, age, sex, and postmortem interval enables the identification of "general trends" in gene expression patterns for that affected tissue (Lukiw, 2013; Clement et al, 2016). To minimize the variations between tissue and tear fluids

miRNA investigations, 3.5 $\mu$l of 10 nM Cel-miR-39 RNA oligos with 5' phosphate modification was added into each homogenized tissue sample before adding chloroform. The quantity and quality of the extracted miRNAs were determined before cDNA preparation in a BioTek (Synergy H1) microplate reader using Gen5 software.

### TaqMan advanced miRNA assays (single-tube assays)

The assays included poly(A) tailing reaction, adaptor ligation reaction, RT reaction, miR-Amp reaction, and real-time PCR. All the steps were based on manufacture's protocols with slight modifications as required. Briefly, 2 $\mu$l of ~10 ng tissue or tear fluid miRNAs were used as the starting material for poly(A) tailing reaction. For fluid samples (serum/plasma), 2 $\mu$l of eluent is recommended as starting material by manufacturer. However, because of the high yield (~100 ng/$\mu$l), and its variations across the mice and age groups, we used fixed amount of tear fluid miRNAs (10 ng) as starting material. 5 $\mu$l of cDNA product was used for 50 $\mu$l miR-Amp reaction, and it was diluted into1:10 with 0.1× TE buffer for real-time PCR. PCR reaction mix contained 5 $\mu$l TaqMan Fast Advanced Master Mix (2×), 0.5 $\mu$l TaqMan advanced miRNA assay (20×), 2 $\mu$l RNase-free water, and 2.5 $\mu$l diluted miR-Amp reaction product per reaction. After the reaction mix was added to the wells, the plate was centrifuged for 1–2 min at 1,000 rpm (Cat# 75003624, SORVALL ST 16R Centrifuge; Thermo Fisher Scientific). qRT-PCR with an initial enzyme activation step of 95°C for 20 s and followed by 40 amplification cycles of 3 s at 95°C and 30 s at 60°C was undertaken on a 7,500 Fast systems real-time PCR instrument (Applied Biosystems). A minimum of three replicates was taken for each sample and for each miRNA. Cel-miR-39 was used as a reference gene for data normalization as the internal controls recommended by manufacturer for TaqMan advanced miRNA assays have been shown to share conserved seed regions with mRNA targets involved in AD or other diseases (e.g., 16-5p and APP/BACE1). Relative miRNA expression level was determined using comparative cycle threshold (Ct) method at a cut off Ct < 40. The mean ± SD and the coefficient of variation (CV) for the relative amount of target miRNAs in both mice groups were calculated using $2^{-\Delta Ct}$ method. The fold change in expression between mice groups was then determined by dividing mean CV of mice group A by mean CV of mice group B (Schmittgen & Livak, 2008).

### Determine the expression level of two human transgenes and its endogenous mouse genes in APP-PS1 mice

Target genes including amyloid precursor protein (APP/App) and presenilin 1 (PSEN1/Psen1) and reference gene glyceraldehyde-3-phosphate dehydrogenase (Gapdh) were determined at young and old ages for all three mice groups. Primer sequences used for this amplification were given (Table S7). Pooled neocortex–hippocampus and pooled eye tissues were used. Total RNA was extracted using RNeasy Mini Kit (Cat# 74104; QIAGEN) as per the manufacturer's protocol. The quantity and quality of the extracted RNA were determined before cDNA preparation. cDNA synthesis was performed with SuperScript VILO cDNA synthesis kit (Cat# 11754050; Invitrogen) as per the manufacturer's protocol. Synthesized cDNAs were then diluted into a concentration of 5 ng/$\mu$l. In brief, the PCR reaction mix per well consisted of 1 $\mu$l of molecular biology grade

water, 5 μl SYBR select master mix, 1 μl of each forward and reverse primers (10 μM), and 2 μl of diluted cDNA (5 ng/μl). After the reaction mix was added to the wells, the plate was centrifuged for 1–2 min at 1,000 rpm. qRT-PCR with an initial enzyme activation step of 95°C for 10 min and followed by 40 amplification cycles of 15 s at 95°C and 1 min at 60°C was undertaken on a 7,500 Fast systems real-time PCR instrument. All target genes were repeated a minimum of three times on each sample. *Gapdh* was used as a reference gene for data normalization. Relative mRNA expression level was determined as similar to miRNA.

### Determine the disease progression using immunofluorescence staining

To visualize the extent of AD-related neuropathological changes over time, the mid-sagittal brain cross sections of Tg mice (n = 8, 4 mice per age group) and NTg siblings (n = 8, 4 mice per age group) at young and old ages were screened using florescence double immunostaining (Wijesinghe et al, 2023). For this purpose, primary antibodies 6E10 (mouse monoclonal, Cat# 803001; BioLegend) which is a marker for 1–16 amino acid residues of Aβ peptide, and neuroinflammatory marker glial fibrillary acidic protein (GFAP, rabbit poly clonal, Cat# Z0334; Dako), a type III intermediate filament protein, and a marker for astrocytes were used. All fluorescent images were captured using a Zeiss LSM 800 confocal microscope with ZEN 2.6 (blue edition) software.

### Data analysis

Mean CV values of miRNAs expressed in different brain regions (neocortex–hippocampus, olfactory bulb, brainstem, and cerebellum), eye tissues, and tear fluids were compared with two different ages as follows: Tg mice versus NTg siblings; Tg mice versus WT controls; and NTg siblings versus WT controls. To determine miRNAs expression levels with ageing and/or disease progression, mean CV values were compared between young and old within each mice group. According to manufacturer, Tg mice and NTg siblings share similar genetic background except two inserted human transgenes. Thus, we compared NTg sibling versus C57BL/6J WT, which is the most widely used background strain in transgenic mouse models to see the differences in tested miRNAs expression levels between these two controls. To determine relative mRNAs expression levels, mean CV values of genes expressed in neocortex–hippocampus, and eye tissues were compared with two different ages as follows: Tg mice versus NTg siblings and Tg mice versus WT controls. Welch's corrected independent samples $t$ test was used (two-tailed) at a significance level of $P < 0.05$. Differentially expressed miRNAs and mRNAs were defined as those that demonstrated a statistically significant two-fold intergroup difference: $\log_2[\text{FC}]$ >1.0 considered up-regulated and <−1.0 considered down-regulated. SPSS version 25.0 (IBM Corp.) was used for the statistical analysis, and the GraphPad Prism 9 (GraphPad Software Inc.) was used to generate graphs.

## Data Availability Statement

The raw data supporting the conclusions of this article will be made available by the authors upon request.

## Supplementary Information

## Acknowledgements

We sincerely thank the Canadian Institute of Health Research, National Sciences and Engineering Research Council of Canada, and National Institutes of Health-NIA R01 AG061138. We would like to thank our laboratory manager Eleanor To for her timely assistances during the entire experimental processes.

### Author Contributions

P Wijesinghe: conceptualization, data curation, formal analysis, investigation, visualization, methodology, and writing—original draft, review, and editing.
J Xi and J Cui: data curation, investigation, and methodology.
M Campbell: investigation, visualization, and methodology.
W Pham: funding acquisition and writing—review and editing.
JA Matsubara: conceptualization, formal analysis, supervision, funding acquisition, project administration, and writing—review and editing.

### Conflict of Interest Statement

The authors declare that the research was conducted in the absence of any commercial or financial relationship that could be construed as a potential conflict of interest.

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
