## [Reviewer comments · Life Science Alliance]

Life Science Alliance

MicroRNAs in tear fluids predict underlying molecular changes associated with Alzheimer's disease

Printha Wijesinghe, Jeanne Xi, Jing Cui, Matthew Campbell, Wellington Pham, and Joanne Matsubara

DOI: <https://doi.org/10.26508/lsa.202201757>

Corresponding author(s): Joanne Matsubara, The University of British Columbia and Printha Wijesinghe, The University of British Columbia

Review Timeline:

Submission Date:	2022-10-06
Editorial Decision:	2022-11-09
Revision Received:	2023-02-10
Editorial Decision:	2023-02-28
Revision Received:	2023-03-06
Accepted:	2023-03-07

Scientific Editor: Novella Guidi

Transaction Report:

November 9, 2022

Re: Life Science Alliance manuscript #LSA-2022-01757-T

Prof. Joanne A Matsubara
The University of British Columbia
Ophthalmology & Visual Sciences
Vancouver
CANADA

Dear Dr. Matsubara,

Thank you for submitting your manuscript entitled "MicroRNAs in tear fluids predict underlying molecular changes associated with Alzheimer's disease" to Life Science Alliance. The manuscript was assessed by expert reviewers, whose comments are appended to this letter. We invite you to submit a revised manuscript addressing the Reviewer comments.

Thank you for this interesting contribution to Life Science Alliance. We are looking forward to receiving your revised manuscript.

Sincerely,

B. MANUSCRIPT ORGANIZATION AND FORMATTING:

Reviewer #1 (Comments to the Authors (Required)):

In the current manuscript, "MicroRNAs in tear fluids predict underlying molecular changes associated with Alzheimer's disease", Wijesinghe et al investigated relative miRNA expression in transgenic APP-PS1 mice, non-carrier siblings, and wildtype (C57BL/6J) controls at young and old ages. They showed that miRNAs associated with amyloid beta (A β) misfolding (-101a, -15a and -342) and proinflammation (-125b, -146a, and -34a) were significantly upregulated in the tear fluids with disease progression. The authors tried to demonstrate the translational potential of tear fluid miRNAs associated with AD pathogenesis. However, the authors need to improve their experimental findings and revise the manuscript considerably to support their conclusions. The reviewer comments are provided below:

The authors hypothesized that "brain, retina and tear fluids will share similarities in their miRNA expression patterns at different stages of AD progression". But they chose to study and refer to miRNA candidates that were shown to be affected in blood samples from AD patients. Although, some of these miRNAs were de-regulated in brains of AD patients at Braak stage III, the author's findings (quantification of miRNA expression) confined to parts of the brain, i.e., neocortex and hippocampus. It makes me wonder whether the studied miRNA candidates do express at an optimum level in the selected parts of the brain which were taken up for study?

In miRbase, there are over 1000 miRNAs reported and yet the authors chose to selectively investigate the 10 miRNAs (Table 1) in their study. Although, they did refer to a couple of earlier published articles (one being a systematic review of peripheral blood samples from AD patients), no where it is mentioned or even remotely indicates that these 10 miRNAs are the prime candidates for this study under the pathological settings, i.e., AD. This reviewer finds it rather difficult to comprehend due to the lack of clear rationale or the mention of it in the text.

Page 5, line 122-123: "... miRNAs -302c and -653 were not detected at Ct < 35 in the brain regions, eye tissues and tear fluids...". It makes me wonder what are the Ct values for the rest of miRNAs and the internal control, Cel-miR-39-3p (which was spiked-in the sample during preparation prior to qPCR)? It was not clear in the materials and methods, how much of cel-39-3p was added to each sample and in what ratio? Considering the possibility that the levels of candidate miRNAs are relatively lower in tear fluids (compared to the rest), the miRNA spike-in (at a high concentration) seriously impacted the measurement of the candidate miRNAs in terms of their Ct values.

In Fig. 2 and Fig. 3, the authors analyzed the candidate miRNA expression in neocortex & hippocampus, eye tissue and tear fluids from young and old animals separately. It would be much more informative if the authors could provide a comparative analysis (of miRNA expression) between the young and old in respective tissues.

The results and discussion are presented in an atypical manner, and it was very hard to comprehend the message the authors trying to deliver (to the reader) from the respective paragraphs. What's the summary of their findings from each section?

The gist of authors findings relies on a single technique, i.e., miRNA quantification using TaqMan assays, which tend to be biased (in terms of specific miRNA quantification) depending on the overall concentration and relative abundance of miRNAs in a given sample. Therefore, to validate the changes in miRNA expression, the authors should provide a second line of verification for the miRNA pattern between the tissue samples in young and old Tg mice.

Page 5, line 122, reference to Table 1 is missing and Fig. 1 was referred to nowhere in the text.

There were no numbers assigned to the figures at the end of the manuscript. Makes it harder to keep track of the figures (main vs supplemental) with their legends and the corresponding referral in the text.

Reviewer #2 (Comments to the Authors (Required)):

In this study, Wijesinghe et al. set out to identify miRNAs that could serve as biomarkers for diagnosis of Alzheimer's disease. They performed TaqMan qPCR assays to measure relative levels of 8 miRNAs in three tissues: 1) Neocortex & Hippocampus; 2) eye tissue; and 3) tear fluids, the latter measured in the hopes of identifying biomarkers for Alzheimer's disease that could be

probed in a non-invasive manner. In addition, these tissues were collected from three mouse lines: a transgenic line and their non-transgene carrying siblings, as well as wild-type controls.

The authors claim that a series of miRNAs previously implicated with amyloid beta misfolding and proinflammation are upregulated in tear fluids concomitant with disease progression. Overall, I find that the article is not written in the most eloquent and clear manner. For example, a clear description in a few sentences of the experimental design and the choice of the particular transgenic animals were lacking in the main text. As someone non-familiar with mouse work, the lack of critical information made for some hard reading.

Importantly, I do not understand how the main conclusions of the paper hold when considering that the tear fluids of old non-transgenic siblings also have upregulated levels of most of these miRNAs (except miR-374c), see middle panels, figure 5. Are there any known parental effects associated with this transgene? Otherwise, it may well be that these miRNAs are simply upregulated as the animals age, in which case they will constitute poor biomarkers for Alzheimer's disease. Before this point is addressed, and taking into account the translational implications of the study, which may misguide future research and impact human disease diagnostics, I cannot in good conscience recommend the publication of this paper.

Another important point would be to indicate the variation underlying the qPCR data. This should be done by at the very least by including the log₂ fold change standard error. Additionally, the datapoints could ideally be superimposed on the barplots. As is, I do not think the barplots are up to publishing standards, and as the main conclusions of this paper rely on such data, this is important to rectify.

Some minor points to consider:

1. I understand this may be technically challenging, but to exclude the possibility that PCR artifacts are hidden within some of these results, did the authors consider validation of expression of particular miRNAs in tear fluids of old transgenic vs young transgenic mice using an additional non-PCR based technique? This could provide strong support to the main conclusions of this paper.

2. Sentence in lines 76/77 one superfluous "the".

3. The usage of "upregulations" is done incorrectly, for example in lines 144-145 and 159-160:

"On the other hand, in the old Tg mice, all miRNAs showed upregulations in the tear fluids. Among them, miRNAs -101a, -146a and -374c were significantly upregulated."

miRNAs should be considered upregulated only if they are significantly so. Otherwise they should not be considered upregulated. Thus, such phrasing is incorrect. It would be correct to say instead: "On the other hand, in the old Tg mice, all miRNAs showed a trend towards upregulation in tear fluids. Among them, RNAs -101a, -146a and -374c were significantly upregulated." Or simply: "miRNAs -101a, -146a, and 374c were significantly upregulated in tear fluids".

4. Line 174, do you mean Supplemental figure 1a and b? Instead of Sup. Fig. 5a and b?

5. Line 178: mir-342 is not significantly downregulated in ageing and disease progression in the Neocortex and Hippocampus of Tg mice, according to figure 5, upper left panel.

Reviewer #3 (Comments to the Authors (Required)):

This study is interesting in investigating tear EVs for AD diagnostics, however, I will suggest the validation of human patients' samples.

Summary blurb (enter in submission system): A short text summarizing in a single sentence the study (max. 200 characters including spaces). This text is used in conjunction with the titles of papers, hence should be informative and complementary to the title and running title. It should describe the context and significance of the findings for a general readership; it should be written in the present tense and refer to the work in the third person. Author names should not be mentioned.

Response #1

Summary blurb is included in this revised version and it is copied and pasted here:

“This study demonstrated the translational potential of deregulated tear fluids miRNAs associated with amyloid beta production and proinflammation in a transgenic AD mouse model.”

Responses to Reviewers:

Reviewer #1 (Comments to the Authors (Required)):

In the current manuscript, "MicroRNAs in tear fluids predict underlying molecular changes associated with Alzheimer's disease", Wijesinghe et al investigated relative miRNA expression in transgenic APP-PS1 mice, non-carrier siblings, and wildtype (C57BL/6J) controls at young and old ages. They showed that miRNAs associated with amyloid beta (A β) misfolding (-101a, -15a and -342) and proinflammation (-125b, -146a, and -34a) were significantly upregulated in the tear fluids with disease progression. The authors tried to demonstrate the translational potential of tear fluid miRNAs associated with AD pathogenesis. However, the authors need to improve their experimental findings and revise the manuscript considerably to support their conclusions. The reviewer comments are provided below:

The authors hypothesized that "brain, retina and tear fluids will share similarities in their miRNA expression patterns at different stages of AD progression". But they chose to study and refer to miRNA candidates that were shown to be affected in blood samples from AD patients.

Response #1a:

This study intends to show the translational potential of tear fluids miRNAs using a transgenic AD mouse model as a proof of concept. It seems that the introduction section containing blood miRNA findings was confusing and therefore we deleted those statements in this revised version. The miRNAs we selected to study have been reported to be deregulated in tissues and cerebrospinal fluids findings as well as blood.

Although, some of these miRNAs were de-regulated in brains of AD patients at Braak stage III, the author's findings (quantification of miRNA expression) confined to parts of the brain, i.e., neocortex and hippocampus. It makes me wonder whether the studied miRNA candidates do express at an optimum level in the selected parts of the brain which were taken up for study?

Response #1b:

To our knowledge, none of the published single study systematically investigated the miRNAs expression levels in AD-affected different brain areas at different time points. So, we do not know whether the miRNA candidates we studied are optimally expressed in neocortex-hippocampus than other parts. However, our study did look and compared different parts of the brain of a transgenic AD mouse model at two different stages. Since the AD-related neuropathological changes are primarily involved in neocortex-hippocampus region and the disease progress in a topographical pattern (Thal's A β phase and Braak and Braak staging system), miRNA findings obtained for neocortex-hippocampus are primarily focused along with eye and tear fluids findings.

The miRNA findings in other brain regions such as olfactory bulb, brainstem and cerebellum are provided under supplementary materials (Tables S1-4 and Figures S1-4).

In miRbase, there are over 1000 miRNAs reported and yet the authors chose to selectively investigate the 10 miRNAs (Table 1) in their study.

Response #2a

We have clearly described about the problems associated with selecting miRNAs and the purpose of this study under introduction and discussion in this revised version.

In lines 95-98 and copied and pasted here:

“In the past decade, there is an enormous number of studies that have reported deregulated single miRNA or panel of miRNAs in early or severe AD-affected human post-mortem brain specimens; AD patients’ cerebrospinal fluids (CSF), serum and plasma samples; and samples obtained from different transgenic AD animal models or cell lines. However, no consensus has been achieved across the studies.”

In lines 100-102 and copied and pasted here:

“Despite accumulated evidence on miRNA deregulation in AD, lacking standardization in samples, sampling time points and quantification methods will impede the chance to characterize miRNAs as potential biomarkers for AD.”

In lines 208-221 and copied and pasted here:

“In the literatures, there exists inconsistencies with respect to deregulated miRNAs associated with AD, making it difficult to use miRNAs as potential diagnostic and prognostic biomarkers. There are several reasons for such discrepancies: variations in types of biological samples used (*e.g.*, different brain regions, CSF, serum, plasma, blood and other body fluids); variations in sampling time points (*e.g.*, early, mid and later stages of the disease progression); variations in transgenic AD animal models and their controls (*e.g.*, APP-PSEN1 mice, 5XFAD mice, 3XTG-AD mice, C57BL/6J AD mice); variations in techniques used to determine the level of expression; and methods used to analyze the data could substantially affect the translational potential of these miRNAs as biomarkers. There are two different APP-PS1 double transgenic mice: B6;C3-Tg (strain#034829-JAX) and B6.Cg-Tg (strain#034832-JAX) that have been widely studied for miRNA investigations in AD. For B6;C3-Tg mice, its noncarrier siblings are recommended as suggested controls. For B6.Cg-Tg mice, its noncarrier siblings are suggested controls and C57BL/6J mice are approximate controls. However, in the published literatures, strain/ control information are generally incomplete. Therefore, in this study, we systematically investigated the miRNAs expression levels in tissue and tear fluids samples using age and sex matched APP-PS1 mice (B6;C3-Tg), its noncarrier siblings and C57BL/6J mice which is the most widely used background strain.”

Although, they did refer to a couple of earlier published articles (one being a systematic review of peripheral blood samples from AD patients), no where it is mentioned or even remotely indicates that these 10 miRNAs are the prime candidates for this study under the pathological settings, i.e., AD. This reviewer finds it rather difficult to comprehend due to the lack of clear rationale or the mention of it in the text.

Response #2b:

In this revised version, we have described about the selection of miRNAs in detail under methodology.

Our selection was primarily based on relevant literature and TargetScan search and described in lines 382-403 and copied and pasted here:

“To test the study hypothesis across brain, eye and tear fluids samples, 10 miRNA candidates were selected including 6 already reported miRNAs -101a-3p, -125b-5p, 146a-5p, -15a-5p, -34a-5p and -342-3p and 4 novel candidates -140-3p, -302c-3p, -374c-5p and -653-5p at the time of study conceptualization (Table S6). Our selection was primarily based on relevant literature and TargetScan search. Anti-amyloidogenic miR-101a (Vilardo et al, 2010, Long and Lahiri, 2011; Li et al, 2019) and -15a, which family miRNAs are anti-amyloidogenic (Liu et al, 2019; Zhang et al, 2020) were selected. Additionally, 3 mostly reported proinflammatory miRNAs -125b, -146a and -34a (Zhao et al, 2015a; Bhattacharjee 2016; Pogue and Lukiw, 2018) were included. Further, miR-342-3p which showed consistent upregulation in the whole brain samples of APP-PS1 mice at 1-, 3-, 6-, and 9- months (Wang et al, 2017) was selected. For TargetScanMouse 7.2 (Agarwal et al, 2015) search, human genes associated with AD were used to identify their biological target miRNAs by searching for the presence of conserved sequences (8mer and 7mer) that match the seed region of each miRNA. miRNAs that shared poorly conserved sequences were excluded. Based on TargetScan search, miRNAs -140-3p which is abundant in the brain (Shao et al, 2010) and shared conserved seed regions with *BACE1*, *BCL2* and *SIRT1*; -302c-3p which is one of the representative large functionally related cluster miRNAs -302a/b/c/d/e, -520a/b/c/d/e/f, -372-3p, and -373-3p those shared conserved seed regions with *APP* and *PSEN1*; -374c-5p which shared conserved seed regions with many AD associated genes including *APP*, *BACE1*, *PSEN1*, *CACNA1C*, *BCL2* and *ATG12* and -653 which is functionally related to another proinflammatory miR-155-5p (Zhao et al, 2015a; Pogue and Lukiw, 2018) based on conserved seed regions with *CACNA1C* and *SIRT1* were selected.”

Page 5, line 122-123: "... miRNAs -302c and -653 were not detected at Ct < 35 in the brain regions, eye tissues and tear fluids...". It makes me wonder what are the Ct values for the rest of miRNAs and the internal control, Cel-miR-39-3p (which was spiked-in the sample during preparation prior to qPCR)?

Response #3a:

We revised the data analysis based on Reviewer 2's suggestion with an extended cut off Ct < 40. Revised data analysis is given in lines 438-445 and copied and pasted here:

“*Cel-miR-39* was used as a reference gene for data normalization as the internal controls recommended by manufacturer for TaqMan advanced miRNA assays have been shown to share conserved seed regions with mRNA targets involved in AD or other diseases (e.g., 16-5p and *APP/ BACE1*). Relative miRNA expression level was determined using comparative cycle threshold (Ct) method at a cut off Ct < 40. The mean \pm standard deviation and the coefficient of variation (CV) for the relative amount of target miRNAs in both mice groups were calculated using $2^{-\Delta Ct}$ method. The fold change in expression between mice groups was then determined by dividing mean CV of mice group A by mean CV of mice group B (Schmittgen and Livak 2008).”

Though we expanded the cut off Ct from 35 to 40, miRNAs- 302c and -653 expressions were undetermined in the tested samples of all 3 mice groups. For other miRNAs, Ct values were <35 across the samples tested and mice groups investigated. Ct values slightly high for miRNA Average Ct values of the internal control, *Cel-miR-39-3p* for tear fluids, eye tissues and brain samples are approximately 7, 11 and 15, respectively.

Revised results showing miRNAs relative expression levels were given in Figures 3-6 and Tables 2-5 for neocortex-hippocampus, eye and tear fluids samples. Revised results showing miRNAs relative expression levels in other brain regions such as olfactory bulb, brainstem and cerebellum were given in Suppl. Figures 1-4 and Suppl. Tables 1-4.

It was not clear in the materials and methods, how much of cel-39-3p was added to each sample and in what ratio?

Response #3b:

We have included those details in this revised version. We added internal control after homogenization during the extraction step.

In lines 404-411 and copied and pasted here:

“200 μ l of supernatant was used as the starting material and 5 volumes QIAzol lysis reagent containing 1.25 μ l bacteriophage MS2 RNA (Cat# 10165948001 Roche, Sigma-Aldrich) as per manufacturer's recommendation for fluids was added to each sample before homogenization. Content was vortexed for 1-2 minutes and the homogenate was incubated at room temperature for 5 minutes, followed by internal control, 3.5 μ l of 10 nM *Cel-miR-39* RNA oligos (Cat# 10620310, Invitrogen) with 5' phosphate modification was added to each sample. Thereafter, an equal volume of chloroform (200 μ l) was added, vortexed for 1 minute and incubated at room temperature for 2-3 minutes. Remaining steps were performed in accordance with manufacturer's instructions. miRNA extraction was individually carried out for each mouse's tear fluids.”

In lines 420-422 and copied and pasted here:

“To minimize the variations between tissue and tear fluids miRNA investigations, 3.5 μ l of 10 nM *Cel-miR-39* RNA oligos with 5' phosphate modification was added into each homogenized tissue sample before adding chloroform.”

Considering the possibility that the levels of candidate miRNAs are relatively lower in tear fluids (compared to the rest), the miRNA spike-in (at a high concentration) seriously impacted the measurement of the candidate miRNAs in terms of their Ct values.

Response #3c:

Our optimized protocol for tear fluids yielded approximately 1.0-1.5 μ g miRNAs from 200 μ l supernatant. Also, we used a fixed amount of tissue or tear fluids miRNAs (approximately 10ng) as starting material to carry out TaqMan advanced miRNA assays and it was described in lines 428-432 and copied and pasted here.

“Briefly, 2 μ l of approximately 10 ng tissue or tear fluids miRNAs were used as the starting material for poly(A) tailing reaction. For fluid samples (serum/plasma), 2 μ l of eluent is recommended as starting material by manufacturer. However, due to the high yield (approximately 100 ng / μ l), and its variations across the mice and age groups, we used fixed amount of tear fluids miRNAs (10 ng) as starting material.”

For example:

Ct value for miR-125b (brain-enriched miRNA) in tear fluids is ~ 22-23, and in neocortex-hippocampus and eye tissues is ~15-18. Ct value for miR-374 (novel candidate miRNA) in tear fluids is ~30-32, and in neocortex-hippocampus and eye tissues is ~25-28.

In Fig. 2 and Fig. 3, the authors analyzed the candidate miRNA expression in neocortex & hippocampus, eye tissue and tear fluids from young and old animals separately. It would be much more informative if the authors could provide a comparative analysis (of miRNA expression) between the young and old in respective tissues.

Response #4a:

In this version, based on reviewers' suggestions, we have revised the figures for each comparison. Further, figures are rearranged/ renamed from the initial submitted version.

Original Suppl. Fig 1 is now Fig. 1

We inserted a new Fig. 2

Original Fig. 1 is now Fig. 8

Original Fig. 2 is now Fig. 4

Original Fig. 3 is now Fig. 5

Original Fig. 4 is now Fig. 3

Original Fig. 5 is now Fig. 6

Original Fig. 6 is now Fig. 7

Original Suppl. Fig 2 is now Fig S2

Original Suppl. Fig 3 is now Fig S3

Original Suppl. Fig 4 is now Fig S1

Original Suppl. Fig 5 is now Fig S4

The results and discussion are presented in an atypical manner, and it was very hard to comprehend the message the authors trying to deliver (to the reader) from the respective paragraphs. What's the summary of their findings from each section?

Response # 5:

We have substantially revised texts under results and discussion in this submission.

We summarized results for the differentially expressed miRNAs based on each comparison and with ageing and/ or disease progression for neocortex-hippocampus, eye tissues and tear fluids. miRNAs relative expression levels in other brain regions were given under supplementary materials.

Discussion was revised to support the major findings based on literature or bioinformatic database (TargetScan) based evidence.

The gist of authors findings relies on a single technique, i.e., miRNA quantification using TaqMan assays, which tend to be biased (in terms of specific miRNA quantification) depending on the overall concentration and relative abundance of miRNAs in a given sample. Therefore, to validate the changes in miRNA expression, the authors should provide a second line of verification for the miRNA pattern between the tissue samples in young and old Tg mice.

Response # 6:

It is true that we did not use a second line of verification in this study. However, to test the hypothesis "that brain (neocortex-hippocampus), eye (retina) and tear fluids will share similarities in their miRNA's expression levels at different stages of AD progression", 6 miRNA candidates were selected from already published original investigations. Further, using TaqMan advanced miRNA assays (single-tube assays) with fixed amount of starting material and internal control across the samples and mice groups, we minimized the potential assay-based bias substantially.

Further, we determined the miRNAs relative expression levels in transgenic APP-PS1 mice by comparing it with its non-carrier siblings and C57BL/6J which is the most widely used background strain. Our findings are consistent with already published original investigations.

Also, according to miRTarBase (the experimentally validated microRNA-target interaction database), qPCR which we used here, is considered as one of the strong evidence or validation methods.

Page 5, line 122, reference to Table 1 is missing and Fig. 1 was referred to nowhere in the text.

Response # 7:

In the original version Table 1 and Fig.1 were given under methodology (following discussion section).

In this revised version, Table 1 is Table S6 and Fig 1 is Fig 8 and given under methodology.

There were no numbers assigned to the figures at the end of the manuscript. Makes it harder to keep track of the figures (main vs supplemental) with their legends and the corresponding referral in the text.

Response #8:

In the original version, we have provided the figure legends (number/caption/title) and supplementary figure legends (number/caption/title) separately in pages 28-30.

Reviewer #2 (Comments to the Authors (Required)):

In this study, Wijesinghe et al. set out to identify miRNAs that could serve as biomarkers for diagnosis of Alzheimer's disease. They performed TaqMan qPCR assays to measure relative levels of 8 miRNAs in three tissues: 1) Neocortex & Hippocampus; 2) eye tissue; and 3) tear fluids, the latter measured in the hopes of identifying biomarkers for Alzheimer's disease that could be probed in a non-invasive manner. In addition, these tissues were collected from three mouse lines: a transgenic line and their non-transgene carrying siblings, as well as wild-type controls.

The authors claim that a series of miRNAs previously implicated with amyloid beta misfolding and proinflammation are upregulated in tear fluids concomitant with disease progression. Overall, I find that the article is not written in the most eloquent and clear manner.

Response #1a:

We have substantially revised manuscript to describe research findings in a clearer manner.

For example, a clear description in a few sentences of the experimental design and the choice of the particular transgenic animals were lacking in the main text. As someone non-familiar with mouse work, the lack of critical information made for some hard reading.

Response #1b

We have included experimental design and choice of the animal model in lines 344-381 and copied and pasted here: "APP/PS1 double transgenic (Tg) mice (B6;C3-Tg, strain#034829-JAX) expressing a chimeric mouse/human amyloid precursor protein (Mo/HuAPP695swe) and a mutant human presenilin 1 (PS1-dE9), its non-carrier siblings (NTg), and its background strain C57BL/6J wildtype (WT) controls were studied. Both mutations are associated with early-onset AD and amyloid plaque formation. Brain, whole eye and tear fluids samples were collected from above 3 mice groups at 2 different ages: 12-16 weeks (young) and 36-40 weeks (old). Since the study was novel and intended to test the hypothesis, a total of 24 female mice were used in 6 groups: young Tg, old Tg, young NTg, old NTg, young WT and old WT (n = 4 mice for each group) using resource equation method (Charan et al, 2013)."

Importantly, I do not understand how the main conclusions of the paper hold when considering that the tear fluids of old non-transgenic siblings also have upregulated levels of most of these miRNAs (except miR-374c), see middle panels, figure 5. Are there any known parental effects associated with this transgene?

Response #2a

It is well documented in the literatures that miRNA can be intragenic, found within the introns of a coding gene or intergenic, found within the introns of another host gene (O'Brien et al., 2018). miRNAs relative expression levels revealed a similar pattern in APP-PS1 (Tg) mice and non-carrier siblings (NTg) when compared with matched C57BL/6J (WT) controls, suggesting that these miRNAs are intergenic. However, the differences seen between Tg mice and NTg siblings could possibly be disease-related intragenic localizations. We summarized the relative expression levels (\log_2 (fold change) with p values) between Tg and NTg in Table 2. All 8 miRNAs showed significant or a trend towards upregulation in the old Tg mice compared with old NTg.

Otherwise, it may well be that these miRNAs are simply upregulated as the animals age, in which case they will constitute poor biomarkers for Alzheimer's disease. Before this point is addressed and taking into account the translational implications of the study, which may misguide future research and impact human disease diagnostics, I cannot in good conscience recommend the publication of this paper.

Response #2b

In this study, to determine the miRNAs relative expressions with ageing and/ or disease progression, we compared young and old mice for all 3 groups. Our findings clearly demonstrated an opposite trend (downregulation in old WT compared to young WT) for C57BL/6J mice with ageing (Table 5). Also, by comparing Tg vs. NTg, we found that

the tested miRNAs relative expression levels showed significant upregulations in old Tg mice compared to old NTg and showed a trend towards upregulation in the tear fluids in young Tg mice compared with young NTg (Table 2). Fig 6C clearly shows the differences in tested miRNAs relative expression levels across the all mice groups for tear fluids, supporting our main conclusion.

Another important point would be to indicate the variation underlying the qPCR data. This should be done by at the very least by including the log2 fold change standard error. Additionally, the datapoints could ideally be superimposed on the barplots. As is, I do not think the barplots are up to publishing standards, and as the main conclusions of this paper rely on such data, this is important to rectify.

Response #3a

We have revised qPCR data as recommended by the reviewer to illustrate relative expression levels across the samples for each comparison group (Figs 3-6, Figs S1-4). Revised data analysis is given in lines 470-474 and copied and pasted here:

“*Cel-miR-39* was used as a reference gene for data normalization as the internal controls recommended by manufacturer for TaqMan advanced miRNA assays have been shown to share conserved seed regions with mRNA targets involved in AD or other diseases (e.g., 16-5p and *APP/ BACE1*). Relative miRNA expression level was determined using comparative cycle threshold (Ct) method at a cut off Ct < 40. The mean ± standard deviation and the coefficient of variation (CV) for the relative amount of target miRNAs in both mice groups were calculated using $2^{-\Delta Ct}$ method. The fold change in expression between mice groups was then determined by dividing mean CV of mice group A by mean CV of mice group B (Schmittgen and Livak 2008).”

Some minor points to consider:

1. I understand this may be technically challenging, but to exclude the possibility that PCR artifacts are hidden within some of these results, did the authors consider validation of expression of particular miRNAs in tear fluids of old transgenic vs young transgenic mice using an additional non-PCR based technique? This could provide strong support to the main conclusions of this paper.

Response 1:

It is true that we did not use a second line of verification in this study. However, to test the hypothesis “that brain (neocortex-hippocampus), eye (retina) and tear fluids will share similarities in their miRNA’s expression levels at different stages of AD progression”, 6 miRNA candidates were selected from already published original investigations. Further, using TaqMan advanced miRNA assays (single-tube assays) with fixed amount of starting material and internal control across the samples and mice groups, we minimized the potential assay-based bias substantially.

Further, we determined the miRNAs relative expression levels in transgenic APP-PS1 mice by comparing it with its non-carrier siblings and C57BL/6J which is the most widely used background strain. Our findings are consistent with already published original investigations.

Also, according to miRTarBase (the experimentally validated microRNA-target interaction database), qPCR which we used here, is considered as one of the strong evidence or validation methods.

2. Sentence in lines 76/77 one superfluous "the".

Response 2:

We have revised the manuscript substantially and corrected grammatical errors as much as possible.

3. The usage of "upregulations" is done incorrectly, for example in lines 144-145 and 159-160:

"On the other hand, in the old Tg mice, all miRNAs showed upregulations in the tear fluids. Among them, miRNAs -101a, -146a and -374c were significantly upregulated."

miRNAs should be considered upregulated only if they are significantly so. Otherwise they should not be considered upregulated. Thus, such phrasing is incorrect. It would be correct to say instead: "On the other hand, in the old Tg mice, all miRNAs showed a trend towards upregulation in tear fluids. Among them, RNAs -101a, -146a and -374c were significantly upregulated." Or simply: "miRNAs -101a, -146a, and 374c were significantly upregulated in tear fluids".

Agreed.

In this revised version, we have used the recommended wordings throughout the main text.

4. Line 174, do you mean Supplemental figure 1a and b? Instead of Sup. Fig. 5a and b?

Response#4

Agreed.

Suppl. Fig. 1 is now Fig 1 and referred appropriately.

5. Line 178: mir-342 is not significantly downregulated in ageing and disease progression in the Neocortex and Hippocampus of Tg mice, according to figure 5, upper left panel.

Response#5

In this revised version, figures (Figures 3-6 and Figures S1-4) were revised to show miRNAs relative expression levels using column graphs (mean \pm S.D) for brain samples and box plots (min to max) for eye tissues and tear fluids.

Reviewer #3 (Comments to the Authors (Required)):

This study is interesting in investigating tear EVs for AD diagnostics, however, I will suggest the validation of human patients' samples.

Response #1

Agreed. However, this study was primarily intended to show the **translational potential of tear fluids microRNAs** using a transgenic AD mouse model as **a proof of concept**. We can prove it only by checking the brain, particularly neocortex-hippocampus in which AD neuropathological changes primarily involved, eye and tear fluids at different time points by using a transgenic AD animal model. This study will be expanded in future to test the tear fluids with additional potential miRNA candidates using different transgenic animal models (*e.g.*, 5xFAD) and at different time points. Upon completion of transgenic AD animal model studies, validation on human tear fluids will be carried out with the approval from research ethics board to conduct a clinical trial in human participants.

February 28, 2023

RE: Life Science Alliance Manuscript #LSA-2022-01757-TR

Prof. Joanne A Matsubara
The University of British Columbia
Ophthalmology & Visual Sciences
Eye Care Centre
Vancouver V5Z 3N9
Canada

Dear Dr. Matsubara,

Thank you for submitting your revised manuscript entitled "MicroRNAs in tear fluids predict underlying molecular changes associated with Alzheimer's disease". We would be happy to publish your paper in Life Science Alliance pending final revisions necessary to meet our formatting guidelines.

- please address the remaining Reviewer 2's concerns
- please add ORCID ID for 1st corresponding author-you should have received instructions on how to do so
- please add the Twitter handle of your host institute/organization as well as your own or/and one of the authors in our system

Figure Check:

- Figure 8 needs a scale bar

A. FINAL FILES:

B. MANUSCRIPT ORGANIZATION AND FORMATTING:

Sincerely,

Reviewer #1 (Comments to the Authors (Required)):

The authors adequately addressed all of this reviewer's concerns and thus, recommended for publication. Thank you

Reviewer #2 (Comments to the Authors (Required)):

The revised manuscript submitted by Wijesinghe and colleagues is significantly improved. The authors have adequately addressed my main concerns regarding the quality of the figures and transparency of data reporting (i.e. barplots and representation of variation).

However, a major concern of mine remains, namely my initial question that led to the author's response 2a and 2b. I understand that miRNAs can originate from either an intergenic or intragenic, typically intronic, locus. However, what I do not understand by the author's explanation is how the miRNA origin relates to the transgenes and the experiments performed.

The sentence starting in line 157: "Finally, to determine whether the tested miRNAs were intergenic or intragenic, expression levels of these miRNAs in neocortex-hippocampus (Fig 5A), eye tissues (Fig 5B) and tear fluids (Fig 5C) were compared between NTg siblings and WT controls". How do the results from Figure 5 inform on the genomic location of a specific miRNA? The location of a miRNA in the genome is a known factor, so the phrasing of these aspects by the authors is highly confusing and imprecise. The experiment in no way informs on the intergenic or intragenic locations of the miRNAs. Considering this is also mentioned in the abstract ("...suggesting that these miRNAs are intergenic. However, the differences seen between APP-PS1 mice and non-carrier siblings could possibly be disease-related intragenic localizations"), it is most important to clarify these points.

A few other minor aspects to consider:

-Panels 2E and G referenced in lines 126 and 127 are not present in figure 2.

Reviewer #2 (Comments to the Authors (Required)):

The revised manuscript submitted by Wijesinghe and colleagues is significantly improved. The authors have adequately addressed my main concerns regarding the quality of the figures and transparency of data reporting (i.e. barplots and representation of variation).

However, a major concern of mine remains, namely my initial question that led to the author's response 2a and 2b. I understand that miRNAs can originate from either an intergenic or intragenic, typically intronic, locus. However, what I do not understand by the author's explanation is how the miRNA origin relates to the transgenes and the experiments performed.

Response #1a

As per manufacturer (JAX laboratory), non-carrier (NTg) siblings are typically made from same genetic background except two inserted genes which express a chimeric mouse/human amyloid precursor protein (Mo/HuAPP695swe) and a mutant human presenilin 1 (PS1-dE9). Our gene expression data clearly demonstrated the differences between Tg mice vs NTg siblings based on human *PSEN1* and mouse *App* mRNA levels in both neocortex-hippocampus and eye tissues at two different time points, young and old ages (Fig 2). **However, since the subsequent upstream or downstream effects of those two inserted human genes in Tg mice are not assured**, we have removed such wordings “intragenic” and “intergenic” in this revised version.

The sentence starting in line 157: "Finally, to determine whether the tested miRNAs were intergenic or intragenic, expression levels of these miRNAs in neocortex-hippocampus (Fig 5A), eye tissues (Fig 5B) and tear fluids (Fig 5C) were compared between NTg siblings and WT controls".

How do the results from Figure 5 inform on the genomic location of a specific miRNA? The location of a miRNA in the genome is a known factor, so the phrasing of these aspects by the authors is highly confusing and imprecise. The experiment in no way informs on the intergenic or intragenic locations of the miRNAs. Considering this is also mentioned in the abstract ("...suggesting that these miRNAs are intergenic. However, the differences seen between APP-PS1 mice and non-carrier siblings could possibly be disease-related intragenic localizations"), it is most important to clarify these points.

Response #1b

Agreed and we explained about it in Response #1a. Revisions made in this submission are given below.

Under abstract, it is revised in lines 42-45 and copied and pasted here:

“Relative expression levels of tested miRNAs revealed a similar pattern in both APP-PS1 mice and non-carrier siblings when compared with age and sex matched wildtype controls. However, the differences seen in expression levels between APP-PS1 mice and non-carrier siblings could possibly have resulted from underlying molecular etiology of AD.”

Under results, it is revised in lines 158-163 and copied and pasted here:

“*NTg siblings vs. WT controls:* As per manufacturer, NTg siblings are typically made from same genetic background except two inserted transgenes which express a chimeric mouse/human amyloid precursor protein (Mo/HuAPP695swe) and a mutant human presenilin 1 (PS1-dE9) in Tg mice. Thus, we compared NTg sibling vs. C57BL/6J WT control, which is the most widely used background strain in transgenic mouse models to see the difference between these two controls. Tested miRNAs’ relative expression levels were illustrated for neocortex-hippocampus (Fig 5A), eye tissues (Fig 5B) and tear fluids (Fig 5C) (Table 4).”

Under discussion, it is revised in lines 240-245 and copied and pasted here:

Although, Tg mice share similar genetic background with NTg siblings except two inserted human transgenes, the subsequent upstream or downstream effects of those two inserted human transgenes in Tg mice are not assured. Therefore, by comparing Tg mice vs. NTg siblings which could possibly have resulted from underlying molecular etiology of AD, the translational potential of tear fluid miRNAs based on its relative expression level, direction of regulation with disease progression and their similarities to neocortex-hippocampus were explored.

Under methods, it is revised in lines 485-488 and copied and pasted here:

“According to manufacturer, Tg mice and NTg siblings share similar genetic background except two inserted human transgenes. Therefore, we compared NTg sibling vs. C57BL/6J WT, which is the most widely used background strain in transgenic mouse models to see the differences in tested miRNAs expression levels between these two controls.”

A few other minor aspects to consider:

-Panels 2E and G referenced in lines 126 and 127 are not present in figure 2.

Response #3

Panels 2E and G are presented in Figure 2. Panel 2E shows the human *APP* and *PSENI* relative expression levels in the eye tissues of Tg mice and WT controls at young and old ages. Panel 2G shows the human *APP* and *PSENI* relative expression levels in the neocortex-hippocampus of Tg mice and WT controls at young and old ages. However, in this revised version, we reorganized column graphs to match the comparison.

Original NTg young vs. Tg young is now Tg young vs. NTg young

Original NTg old vs. Tg old is now Tg old vs. NTg old

Original WT young vs. Tg young is now Tg young vs. WT young

Original WT old vs. Tg old is now Tg old vs. WT old

March 7, 2023

RE: Life Science Alliance Manuscript #LSA-2022-01757-TRR

Prof. Joanne A Matsubara
The University of British Columbia
Ophthalmology & Visual Sciences
Eye Care Centre
Vancouver V5Z 3N9
Canada

Dear Dr. Matsubara,

Thank you for submitting your Research Article entitled "MicroRNAs in tear fluids predict underlying molecular changes associated with Alzheimer's disease". It is a pleasure to let you know that your manuscript is now accepted for publication in Life Science Alliance. Congratulations on this interesting work.

DISTRIBUTION OF MATERIALS:

Again, congratulations on a very nice paper. I hope you found the review process to be constructive and are pleased with how the manuscript was handled editorially. We look forward to future exciting submissions from your lab.

Sincerely,
